# Semisynthetic biosensors for mapping cellular concentrations of nicotinamide adenine dinucleotides

Olivier Sallin[1], Luc Reymond[1,2], Corentin Gondrand[3], Fabio Raith[3], Birgit Koch[3], Kai Johnsson[1,3,2]*

[1]École Polytechnique Fédérale de Lausanne, Institute of Chemical Sciences and Engineering, Lausanne, Switzerland; [2]National Centre of Competence in Research in Chemical Biology, Lausanne, Switzerland; [3]Department of Chemical Biology, Max-Planck-Institute for Medical Research, Heidelberg, Germany

**Abstract** We introduce a new class of semisynthetic fluorescent biosensors for the quantification of free nicotinamide adenine dinucleotide ($NAD^+$) and ratios of reduced to oxidized nicotinamide adenine dinucleotide phosphate ($NADPH/NADP^+$) in live cells. Sensing is based on controlling the spatial proximity of two synthetic fluorophores by binding of NAD(P) to the protein component of the sensor. The sensors possess a large dynamic range, can be excited at long wavelengths, are pH-insensitive, have tunable response range and can be localized in different organelles. Ratios of free $NADPH/NADP^+$ are found to be higher in mitochondria compared to those found in the nucleus and the cytosol. By recording free $NADPH/NADP^+$ ratios in response to changes in environmental conditions, we observe how cells can react to such changes by adapting metabolic fluxes. Finally, we demonstrate how a comparison of the effect of drugs on cellular NAD(P) levels can be used to probe mechanisms of action.

DOI: https://doi.org/10.7554/eLife.32638.001

*For correspondence: johnsson@mr.mpg.de

## Introduction

Nicotinamide adenine dinucleotide (NAD) and its phosphorylated form NADP are cofactors involved in a multitude of redox reactions regulating energy metabolism, reductive biosynthesis and antioxidant defense. $NAD^+$ is also a cofactor for sirtuins and poly(ADP-ribose) polymerases (PARPs), enzymes which regulate numerous important cellular functions (*Cantó et al., 2015*; *Verdin, 2015*; *Peek et al., 2013*). Due to the central role of NAD(P) in various biological processes and multiple pathologies (*Cantó et al., 2015*; *Verdin, 2015*), the quantification of their concentrations is of great importance.

NAD(P) is compartmentalized and present as free and protein-bound fractions within cells. Different methods are currently used to quantify total NAD(P) concentrations and their ratios in cell extracts (*Yang et al., 2007*; *Lowry et al., 1961*; *Vidugiriene et al., 2014*). However, the results obtained by these methods have limited physiological relevance because the majority of pyridine nucleotides is known to be protein-bound (*Zhang et al., 2002*) and have different distribution between cytosol and mitochondria (*Williamson et al., 1967*). Free $NAD(P)H/NAD(P)^+$ ratios can be indirectly determined by measuring the ratio of selected redox couples (*Williamson et al., 1967*; *Veech et al., 1969*). Yet, such approaches lack spatial resolution and are not suitable for studying dynamic changes. Several genetically encoded fluorescent sensors have been developed to study the spatiotemporal dynamics of these cofactors. Current sensors can measure changes in free $NAD^+/NADH$ ratio (*Zhao et al., 2015*), NADH (*Zhao et al., 2011*; *Hung et al., 2011*), $NAD^+$ (*Cambronne et al., 2016*), $NADP^+$ (*Cameron et al., 2016*) as well as NADPH (*Tao et al., 2017*).

SoNar and iNAP, two fluorescent sensors for measuring free $NAD^+$/NADH and NADPH, respectively, are particularly well performing NAD(P) sensors as they are bright, ratiometric and show a large dynamic range. SoNar and iNAP are based on inserting cpYFP into the redox-sensing transcriptional repressor Rex (*Zhao et al., 2015*; *Tao et al., 2017*). However, both sensors require excitation at short wavelengths (420 nm and 480 nm) and the fluorescence signal upon excitation at 480 nm is pH-dependent. A sensor for measuring free $NAD^+$ has been generated by fusing a bipartite $NAD^+$-binding protein to cpVenus (*Cambronne et al., 2016*). While being the first sensor able to measure free, compartmentalized $NAD^+$, it only shows a modest two-fold dynamic range and requires excitation at 405 and 488 nm. Furthermore, the pH sensitivity of the fluorescence signal of the sensor between pH 7.4 to 8 is comparable to its dynamic range. In addition, none of the sensors introduced so far permits a rational adaption of their response range and no sensors exist to measure free NADPH/$NADP^+$. Consequently, additional sensors measuring cellular levels of NAD(P) are needed to study their role in metabolism and signaling.

Here, we introduce a new class of semisynthetic fluorescent biosensors for measuring cellular free $NAD^+$ and NADPH/$NADP^+$. The sensors are ratiometric, display large dynamic ranges, are pH-insensitive, possess tunable response range and can be excited at long wavelengths (560 nm). Together, these properties make them powerful tools for mapping temporal dynamics of cellular concentrations of NAD(P).

## Results

### Sensor design and characterization

Our NAD(P) sensor design is based on the Snifit concept (*Brun et al., 2009*). Snifits contain an analyte-binding protein and two self-labeling protein tags, for example SNAP-tag (*Keppler et al., 2003*) and Halo-tag (*Los et al., 2008*). The tags permit the site-specific attachment of two synthetic fluorescent probes, whereas one of the probes also comprises a ligand for the receptor. Analyte binding affects interaction of the tethered ligand with the protein component, thereby affecting the distance between the fluorophores and resulting in FRET efficiency changes. For the design of NAD(P)-Snifits, we selected human sepiapterin reductase (SPR) as NADP-binding protein. As tethered ligand, we focused on sulfa drugs. These potent SPR inhibitors such as sulfapyridine and sulfamethoxazole form a ternary complex with the enzyme in the presence of $NADP^+$, but not with NADPH (*Figure 1a*) (*Chidley et al., 2011*; *Haruki et al., 2012*). We speculated that the π-stacking interaction between the sulfa drug and the nicotinamide moiety of $NADP^+$ could be exploited to generate a semisynthetic biosensor for $NADP^+$ (*Figure 1a,b*). The designed sensor (termed NADP-Snifit) is a fusion protein containing SPR, SNAP-tag and Halo-tag. SNAP-tag is labeled with a molecule (CP-TMR-SMX) that contains sulfamethoxazole as ligand and a tetramethylrhodamine derivative (TMR) as fluorophore. Halo-tag is labeled with SiR-Halo, a siliconrhodamine (SiR) derivative that can act as FRET acceptor for TMR (*Figure 1b,c*). According to our design principle, the tethered sulfamethoxazole should bind to SPR in an $NADP^+$-dependent manner, thereby increasing FRET efficiency between the two fluorophores. In the design of CP-TMR-SMX, we attempted to minimize the size of the molecule to ensure cell permeability. The tetramethylrhodamine derivative was therefore integrated in the linker between sulfamethoxazole and the substrate for SNAP-tag (*Figure 1c*). In order to maximize FRET efficiency of the closed state, Halo-tag was fused to the C-terminus of SPR, bringing SiR close to the ligand binding site of SPR. To decrease FRET efficiency of the open state of the sensor, a proline-30 linker was introduced between SNAP-tag and Halo-tag (*Brun et al., 2011*).

Labeling of the fusion protein SPR-Halo-p30-SNAP with CP-TMR-SMX and SiR-Halo was fast, with second-order rate constants of $3.9 \cdot 10^4$ and $2.5 \cdot 10^5$ $M^{-1}s^{-1}$, respectively (*Appendix 1—table 1*). Titration of the resulting NADP-Snifit with $NADP^+$ revealed a maximum 8.9 ± 0.1 fold FRET ratio change (*Figure 2a,b*). The concentration resulting in the half-maximal sensor response ($c_{50}$) was determined to be 29 ± 7 nM (*Figure 2a,b*). No binding of intramolecular ligand was detectable in the absence of $NADP^+$ (*Appendix 1—Figure 1a*). Titration of the sensor with NADPH showed that the intramolecular ligand does not bind to the binary complex of SPR:NADPH, presumably due to the absence of the π-stacking interaction (*Appendix 1—Figure 1b*). As both cofactors compete for the same binding site, the equilibrium between the open and closed state of NADP-Snifit is controlled by the ratio of NADPH/$NADP^+$. Titration of the sensor with varying NADPH/$NADP^+$ ratios

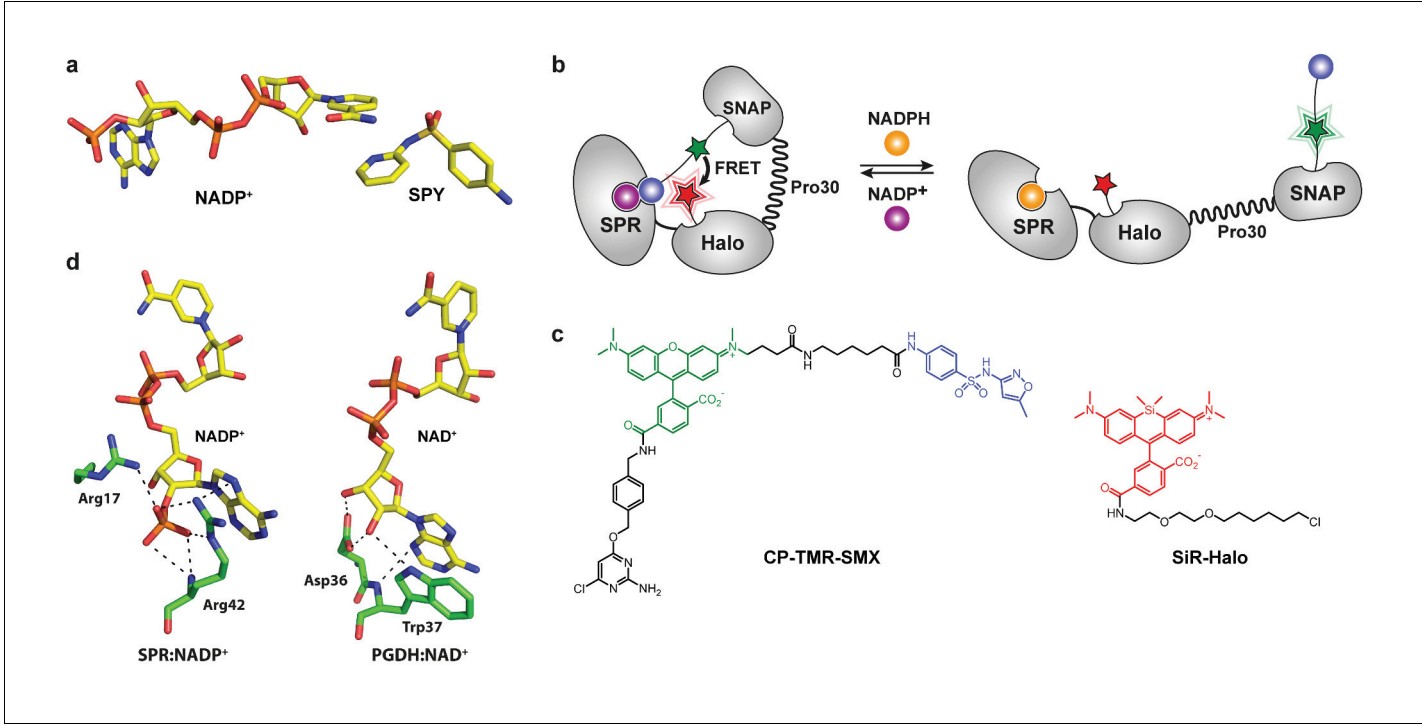

**Figure 1.** Design of semisynthetic sensors for NADP and NAD$^+$. (a) Interaction of NADP$^+$ and sulfapyridine in the substrate-binding site of SPR (PDB entry: 4HWK). The pyridine moiety of sulfapyridine (SPY) and the nicotinamide moiety of NADP$^+$ are at a suitable distance (3.3 Å) for efficient $\pi$-stacking. (b) The fusion protein SPR-Halo-p30-SNAP is labeled via SNAP-tag with a synthetic molecule containing a FRET donor (green star) and a SPR inhibitor (blue ball, SMX), and via Halo-tag with a FRET acceptor. NADPH (orange ball) and NADP$^+$ (purple ball) compete for the cofactor-binding site of SPR. The sensor can monitor NADPH/NADP$^+$ ratio changes by switching from a closed conformation to an open conformation, with high and low FRET efficiency, respectively. (c) Structures of the synthetic molecules used to constitute the sensor. CP-TMR-SMX contains $O^4$-benzyl-2-chloro-6-aminopyrimidine (CP) for reaction with SNAP-tag, a tetramethylrhodamine (TMR, green) derivative as FRET donor and a tethered sulfamethoxazole (SMX, blue). SiR-Halo is used for the specific labeling of Halo-tag with siliconrhodamine. (d) Interactions of residues contributing to cofactor specificity of the SDR superfamily. NADP(H)-preferring enzymes (e.g. SPR) have two conserved basic residues interacting directly with the 2'-phosphate group of NADP$^+$ (PDB entry: 4HWK). NAD(H)-preferring enzymes (e.g. PGDH) have a conserved aspartic acid interacting in a bidentate manner with the 2'- and 3'-hydroxyl groups of NAD$^+$ (PDB entry: 2GDZ).

DOI: https://doi.org/10.7554/eLife.32638.002

showed that the half-maximal sensor response ($r_{50}$) corresponds to a ratio of 30 ± 3 (*Figure 2c*). As cellular free NADPH/NADP$^+$ values have been reported to be between 10 and 100 (*Veech et al., 1969*; *Hedeskov et al., 1987*; *Zhang et al., 2015*), NADP-Snifit in cells would report on free NADPH/NADP$^+$ and not free NADP$^+$.

The modular design of NADP-Snifit permits its redesign into a sensor for NAD$^+$. SPR belongs to the short-chain dehydrogenase/reductase (SDR) superfamily and has a characteristic Rossmann fold

**Table 1.** Quantification of free NADPH/NADP$^+$ and NAD$^+$ levels in different subcellular compartments of U2OS cells.

| | NADPH/NADP$^+$ | | NAD$^+$ (µM) | |
| --- | --- | --- | --- | --- |
| | Emission ratio | TCSPC-FLIM | Emission ratio | TCSPC-FLIM |
| Cytosol | 64.9 ± 26.1 | 55.8 ± 11.7 | 52.8 ± 21.6 | 73.9 ± 7.1 |
| Nucleus | 51.0 ± 16.7 | 40.4 ± 6.7 | n.d. | 117.8 ± 7.2 |
| Mitochondria | 218.7 ± 107.2 | 175.3 ± 57.9 | n.d. | 95.6 ± 7.3 |

The values represent the mean ± s.d. of n = 60 and n = 10 cells for the emission ratio and FLIM measurements, respectively. n.d., not determined.

DOI: https://doi.org/10.7554/eLife.32638.010

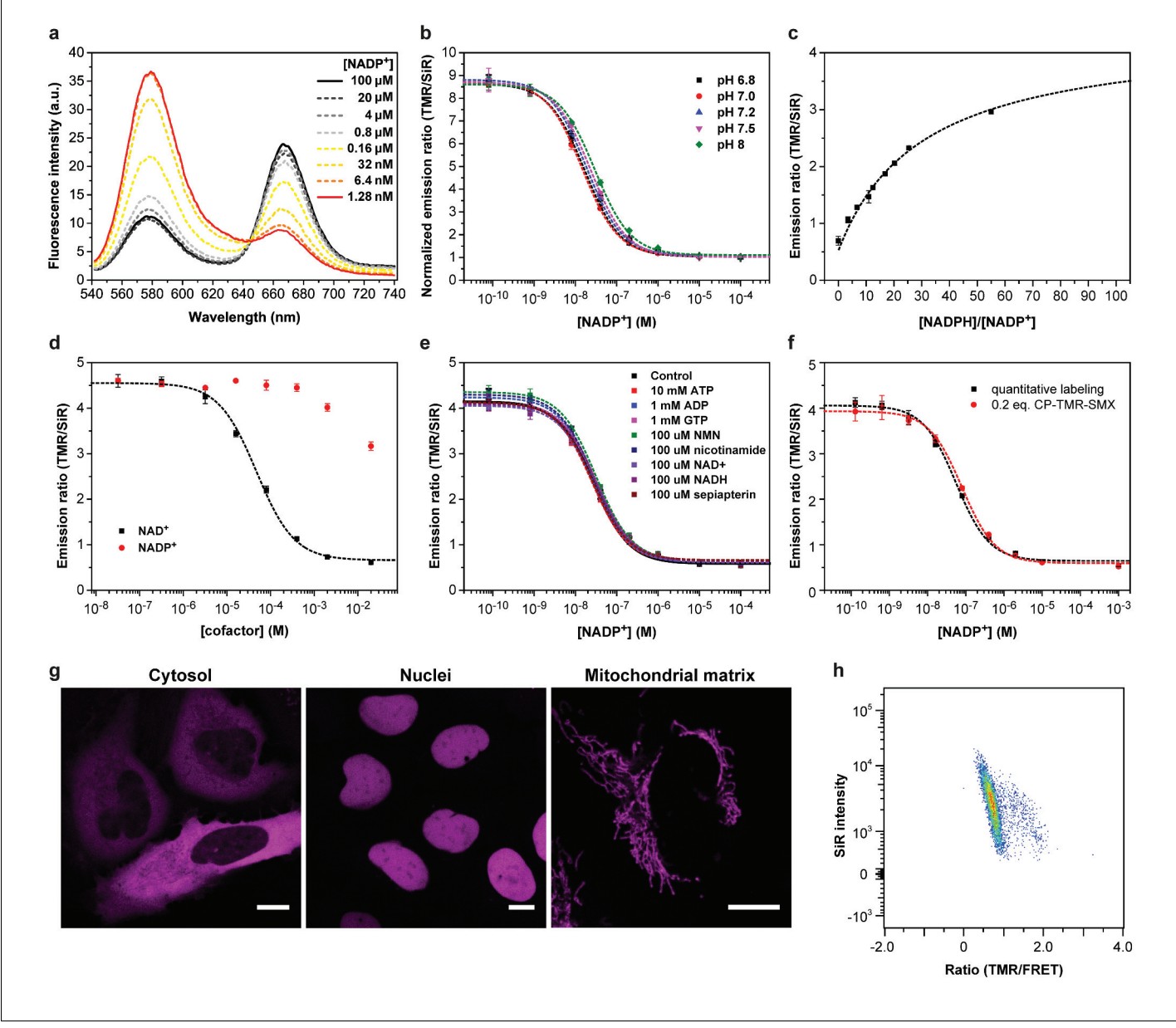

**Figure 2.** Characterization of NADP- and NAD-Snifit. (**a**) Emission spectra of NADP-Snifit titrated with $NADP^+$. TMR and SiR have a maximal emission at 577 and 667 nm, respectively, and the sensor has an isosbestic point at 645 nm. (**b**) Titrations of NADP-Snifit with $NADP^+$ at various pH ranging from 6.8 to 8.0. The maximum FRET ratio change is $8.9 \pm 0.1$ fold with a $c_{50}$ of $29 \pm 7$ nM. (**c**) Titration of NADP-Snifit with NADPH/$NADP^+$. The ratio NADPH/$NADP^+$ corresponding to the half maximal sensor response, $r_{50}$ is $30 \pm 3$. For the fitting, the upper asymptote is set to the value obtained by adding saturating concentration of sulfamethoxazole (2 mM). (**d**) Titration of NAD-Snifit labeled with CP-TMR-SMX and SiR-Halo. The maximum ratio change is $7.6 \pm 0.2$ fold with a $c_{50}$ of $63 \pm 12$ μM. (**e**) Titrations of NADP-Snifit with $NADP^+$ in presence of a fixed concentration of one of the listed different metabolites and structural analogs. (**f**) Comparative titrations between a quantitatively labeled sensor protein and a sensor protein only labeled with 0.2 equivalent of CP-TMR-SMX. The different fitted parameters from the titration and kinetic experiments are obtained from three independent titrations performed in triplicate. Data represent the mean ± s.d. (**g**) Confocal images of U2OS cells expressing NADP-Snifit in defined cellular compartments. The images represent the SiR fluorescence of the labeled sensor. Scale bars, 10 μm. (**h**) Representative gated population of cytosolic NAD-Snifit in U2OS measured by flow cytometry (7000 cells). The graph represents SiR intensity through direct excitation versus FRET ratio.
DOI: https://doi.org/10.7554/eLife.32638.003

The following source data is available for figure 2:

**Source data 1.**
DOI: https://doi.org/10.7554/eLife.32638.004
**Source data 2.**
*Figure 2 continued on next page*

*Figure 2 continued*

DOI: https://doi.org/10.7554/eLife.32638.005

**Source data 3.**

DOI: https://doi.org/10.7554/eLife.32638.006

**Source data 4.**

DOI: https://doi.org/10.7554/eLife.32638.007

**Source data 5.**

DOI: https://doi.org/10.7554/eLife.32638.008

**Source data 6.**

DOI: https://doi.org/10.7554/eLife.32638.009

as dinucleotide-binding domain (*Kallberg et al., 2010*). Enzymes of that superfamily utilize either NAD or NADP as cofactors. Enzymes specific for NADP, such as SPR, generally possess two conserved arginines or lysines interacting with the 2'-phosphate group and the adenine moiety. Enzymes specific for NAD, such as 15-hydroxyprostaglandin dehydrogenase (PGDH), have a conserved aspartate that interacts with 2'- and 3'-hydroxyl groups in a bidentate manner (*Figure 1d*). Guided by sequence and structure comparison of SPR and PGDH (*Tanaka et al., 1996*), we switched the cofactor specificity of NADP-Snift by introducing the mutations A41D and R42W into SPR. Titrations of the resulting NAD-Snifit with either $NAD^+$ or $NADP^+$ showed that the sensor was specific for $NAD^+$ with a $c_{50}$ of $63 \pm 12$ µM while conserving the 8-fold maximum ratio change of NADP-Snifit (*Figure 2d*). NAD-Snifit did not show any response to $NADP^+$ up to concentrations of 1 mM. Under physiological conditions, the reported free cytosolic $NAD^+$ of mammalian cells is around 100 µM (*Zhang et al., 2002*; *Cambronne et al., 2016*) and the $NAD^+/NADH$ ratio has been reported to be 100–600 (*Veech et al., 1969*; *Zhao et al., 2015*) in the cytosol and 4–10 (*Veech et al., 1969*; *Williamson et al., 1967*) in the mitochondria. In cells, NAD-Snifit would thus report on free $NAD^+$ levels.

We tested the interaction of these two sensors with eight key metabolites, including the SPR substrate sepiapterin (*Figure 2e* and *Appendix 1—Figure 1d*). We could not observe any interference at physiologically relevant concentrations of any of these metabolites. In addition, both sensors show negligible pH sensitivity between pH 6.8 and 8 (*Figure 2b* and *Appendix 1—Figure 1c*). While both sensors displayed a two-fold increase of their $c_{50}$ values when raising the temperature from 25°C to 37°C (*Appendix 1—Figure 1e, f*), the $r_{50}$ of NADP-Snifit was not affected by such temperature changes (*Appendix 1—Figure 1g*), indicating that the affinities of SPR for NADPH and $NADP^+$ display similar temperature dependencies.

The opening of closed NADP-Snifit bound with $NADP^+$ follows first-order kinetics with a half-life $t_{1/2}$ of $25 \pm 1$ s, whereas the closing of open sensor upon binding of $NADP^+$ is much faster with a $t_{1/2}$ of <1 s (*Appendix 1—Figure 1h, i*). As NAD-Snifit has a 1000-fold lower affinity for its cofactor than NADP-Snifit, we assume that the kinetics of NAD-Snifit should be at least as fast as those of NADP-Snifit. Accordingly, both NADP-Snifit and NAD-Snifit are suitable to monitor fluctuations of NADPH/$NADP^+$ and $NAD^+$ on the time scale of seconds.

The rational design principle and modular character of the two sensors facilitate the engineering of their properties. For example, the response range of the sensor can be tuned by changing the affinity of the tethered ligand. Replacing the tethered sulfamethoxazole with sulfachloropyridazine, a ligand with lower affinity to SPR, raised the $c_{50}$ of NADP-Snifit from $29 \pm 7$ nM to $1.9 \pm 0.3$ µM (*Appendix 1—figure 1j, k*). The spectral properties of NAD(P)-Snifits can be tuned by simply exchanging the fluorophores: exchanging Halo-tag with EGFP yields a FRET sensor with green excitation maximum, and TMR as FRET acceptor (*Appendix 1—figure 1l–n*).

We then expressed and labeled the sensor in the cytosol, nucleus and mitochondria of different mammalian cells (*Figure 2g* and *Appendix 1—figure 2*). For nuclear and mitochondrial localizations, the Snifits were expressed with appropriate localization sequences. As intracellular labeling is a prerequisite for cellular applications of the sensor, we determined labeling efficiencies in live cells. Intracellular labeling of the sensors with SiR-Halo and CP-TMR-SMX was achieved by simple incubation of the cells with the substrates. The labeling efficiency of SiR-Halo was 100% and of CP-TMR-SMX 92% (*Appendix 1—figure 3a, b*). Despite the incomplete labeling with CP-TMR-SMX, the ratiometric readout can still be used for the quantification of NADPH/$NADP^+$ or $NAD^+$ as there is negligible

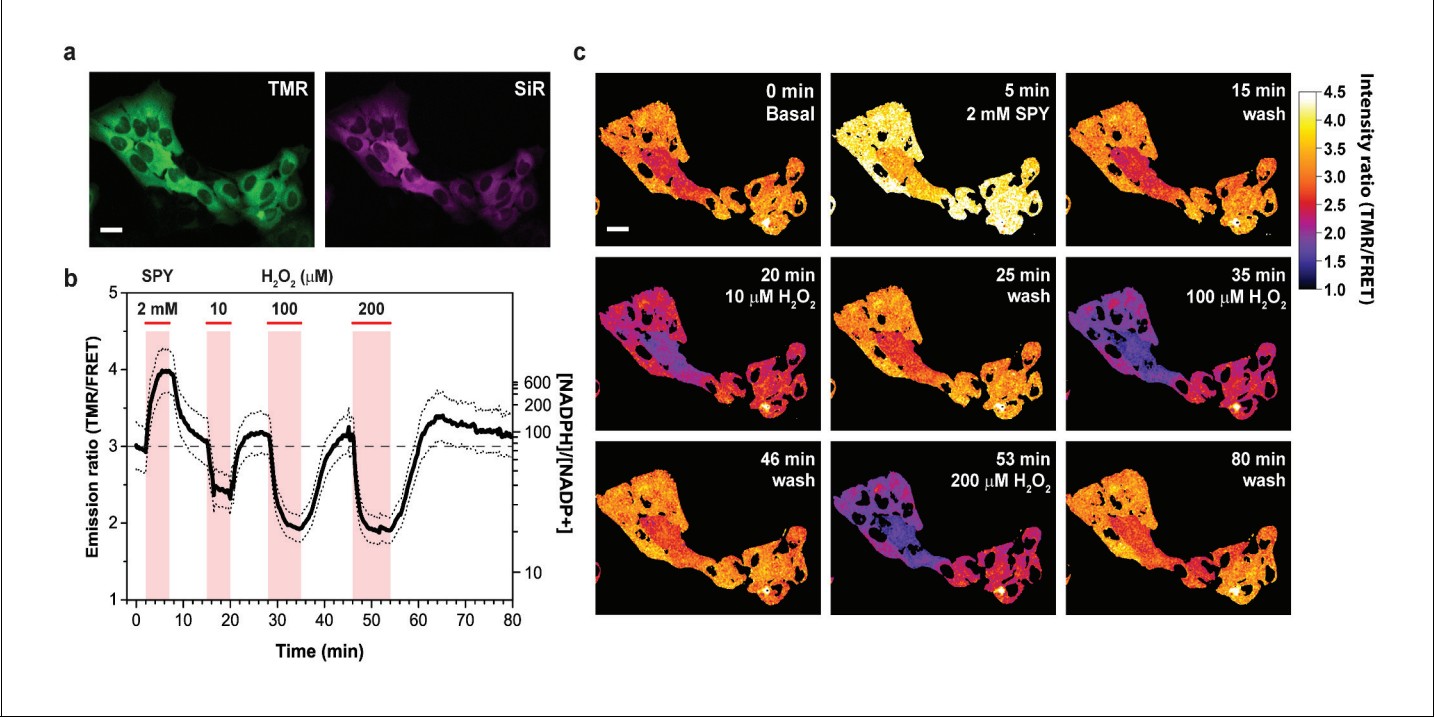

**Figure 3.** Response of cytosolic NADP-Snifit to $H_2O_2$ perfusion. (**a**) Pseudocolored widefield images of cytosolic NADP-Snifit expressed and labeled in U2OS cells corresponding to the donor channel (TMR, green) and acceptor channel through direct excitation (SiR, magenta). (**b**) Time course of the FRET ratio (TMR/FRET) of cytosolic NADP-Snifit upon perfusion of 2 mM sulfapyridine (SPY; to determine the FRET ratio of the sensor in the open state in situ) and increasing concentration of $H_2O_2$ (10, 100, 200 µM). The continuous line represents the mean ratio ±s.d. (dotted lines) (n = 10 cells). Free NADPH/NADP$^+$ ratios are represented on the right y-axis. The red bars indicate the time span of perfusion. (**c**) Ratio images of the cytosolic NADP-Snifit at different time points. Scale bars, 30 µm.

DOI: https://doi.org/10.7554/eLife.32638.011

The following source data is available for figure 3:

**Source data 1.**

DOI: https://doi.org/10.7554/eLife.32638.012

direct excitation of the FRET acceptor. Calibration curves of FRET ratio versus NADP$^+$ of NADP-Snifits, labeled in vitro either with 20% or 100% of CP-TMR-SMX, fully overlay (*Figure 2f*). Furthermore, when using fluorescence-lifetime imaging microscopy (FLIM) for quantification, partial labeling with CP-TMR-SMX does not affect quantifications.

The sensors are also well suited for analysis via flow cytometry (*Figure 2h* and *vide infra*). In such experiments, the FRET ratio was shown to be largely independent of the intensity of the TMR or SiR signals, indicating that neither variations in labeling efficiency nor expression level affect quantitative analysis.

We determined the intracellular sensor concentration reached in different cellular compartments to be in the low micromolar range (1–5 µM) (*Appendix 1—table 2*). Several dehydrogenases are among the most abundant cellular proteins (*Beck et al., 2011*), and there is a large pool of proteins that buffer NAD(P) (*Zhang et al., 2002*). In U2OS cells, SPR itself is a highly abundant protein (*Beck et al., 2011*), and we determined comparable high levels of endogenous SPR in a number of different cell lines (*Appendix 1—figure 3c, d*). Thus, the additional buffering produced by the presence of the sensor protein in the low micromolar range should be negligible.

## Subcellular quantification of free NADPH/NADP$^+$ and NAD$^+$

We quantified free NADPH/NADP$^+$ and NAD$^+$ in different subcellular compartments by time-correlated single photon counting FLIM (TCSPC-FLIM) as the accuracy of FRET measurements by FLIM outperforms other techniques such as two-channel intensity imaging and spectral imaging

**Table 2.** Pharmacological alterations of NAD$^+$ and NADPH/NADP + in U2OS cells measured by flow cytometry.

| | Normalized FRET ratio (TMR/FRET) | | | |
| | NAD-Snifit | | NADP-Snifit | |
| Treatment | Cytosol | Mitochondria | Cytosol | Mitochondria |
| --- | --- | --- | --- | --- |
| Control | 1.00 (±0.03) | 1.00 (±0.02) | 1.00 (±0.01) | 1.00 (±0.01) |
| 1 mM NA | 0.91 (±0.01) | n.d. | 0.92 (±0.01) | 1.00 (±0.01)* |
| 10 mM NAM | 0.92 (±0.02) | n.d. | 1.05 (±0.01) | n.d. |
| 1 mM NMN | 0.82 (±0.01) | n.d. | 0.95 (±0.01) | 0.99 (±0.01)* |
| 1 mM NR | 0.80 (±0.02) | n.d. | 0.96 (±0.01) | 1.00 (±0.01)* |
| 100 nM FK866 | 1.61 (±0.06) | 1.48 (±0.04) | 1.05 (±0.01) | 0.99 (±0.01)* |
| 1 mM 6-AN | n.d. | n.d. | 0.80 (±0.02) | n.d. |
| 1 mM Metformin | 0.89 (±0.04) | 1.09 (±0.03) | 0.90 (±0.01) | 0.95 (±0.01) |
| 1 mM Phenformin | 0.79 (±0.05) | 1.13 (±0.06) | 0.88 (±0.01) | 0.83 (±0.01) |
| 10 μM Rotenone | 0.67 (±0.03) | 1.08 (±0.02) | 0.75 (±0.02) | 0.80 (±0.02) |
| 25 μM Oligomycin A | 1.14 (±0.03) | 1.63 (±0.01) | 1.12 (±0.03) | 1.36 (±0.07) |

Values represent the average of medians (±s.d.) TMR/FRET ratios of three independent measurements normalized to control condition (n = 3). Control: untreated cells (full growth medium with 25 mM glucose), NA: nicotinic acid, NAM: nicotinamide, NMN: nicotinamide mononucleotide, NR: nicotinamide riboside, FK866: (E)-N-[4-(1-benzoylpiperidin-4-yl)butyl]−3-(pyridin-3-yl)acrylamide, 6-AN: 6-aminonicotinamide.

*The effect of the treatment is not statistically significant compared to the control condition (Kruskal-Wallis with Dunn's post-hoc multiple comparison test, α = 0.05). n.d., not determined. All compounds were also tested for interactions with the sensor in vitro (*Appendix 1—figures 5*, *6*).

DOI: https://doi.org/10.7554/eLife.32638.013

(*Pelet et al., 2006*). FRET efficiencies (E) and free NADPH/NADP$^+$ or NAD$^+$ are related by the following equations:

$$\frac{[NADPH]}{[NADP^+]} = K_{50}\frac{E_{max} - E}{E - E_{min}} \tag{1}$$

$$[NAD^+] = K_D^{'}\frac{E - E_{min}}{E_{max} - E} \tag{2}$$

where $E_{max}$ and $E_{min}$ correspond to the maximal and minimal FRET efficiencies, $K_{50}$ is the ratio of NADPH/NADP$^+$ at half saturation and $K_D'$ is the apparent dissociation constant for NAD$^+$. Incubation with 2 mM sulfapyridine allowed us to fully shift the sensor to its open state and to obtain $E_{min}$. Ideally, $E_{max}$, $K_{50}$ and $K_D'$ should be determined in cells. However, concentrations of NAD(P) are difficult to calibrate *in cellulo* due to their cell impermeability and the presence of NAD(P)-dependent enzyme-substrate pairs. Permeabilizing cells with a detergent and equilibrating the cell with an extracellular buffer of known NADP$^+$ concentration (*Zhao et al., 2011*; *Cambronne et al., 2016*) in our hands yielded unreliable results as the sensor diffuses relatively fast out of the cells and digitonin treatment even at low concentrations (0.001%) is toxic. However, the dynamic range of the sensors (e.g. maximum FRET ratio change) in digitonin-permeabilized cells and in cell lysates was identical to the values determined in buffer (*Appendix 1—figure 3e, f*). We therefore used the $E_{max}$, $K_{50}$ and $K_D'$ values determined in vitro for the cellular quantifications.

The free NADPH/NADP$^+$ and NAD$^+$ values of the different cellular organelles in U2OS cells obtained by FLIM are reported in *Table 1*. We also performed a subcellular quantification of free NAD$^+$ and NADPH/NADP$^+$ in U2OS cells by emission ratio imaging (*Table 1*). Furthermore, cytosolic NADPH/NADP$^+$ and NAD$^+$ levels were quantified in NIH/3T3, HeLa and HEK-293T cell lines (*Appendix 1—table 3*).

In our measurements, the values obtained by FLIM and emission ratio imaging agreed very well (*Table 1*). With respect to free $NAD^+$ levels, free intracellular $NAD^+$ in U2OS cells was found to be around 70–120 µM. Free cytosolic $NAD^+$ of the different cell lines were found to be relatively similar, ranging from 40 to 70 µM (*Table 1*, *Appendix 1—table 3*). These results are in agreement with previously reported values for HEK293 cells, obtained with the cpVenus-based $NAD^+$ sensor (*Cambronne et al., 2016*). With respect to $NADPH/NADP^+$, we discovered that free $NADPH/NADP^+$ is maintained at a high ratio inside cells while the reduction potential of mitochondria is significantly higher than that of the cytosol and the nucleus (*Table 1*). Free cytosolic $NADPH/NADP^+$ ratios in the different cell lines varied up to 4-fold, ranging from 20 to 80 (*Appendix 1—table 3*). To our knowledge, it is the first time that free, cellular $NADPH/NADP^+$ is directly quantified and that a difference in this ratio between cytosol and mitochondria is demonstrated. The higher ratio of $NADPH/NADP^+$ in mitochondria could, at least partially, be due to the higher pH in that organelle, pushing mitochondrial NAD(P) transhydrogenase and dehydrogenases towards the formation of NADPH (*Rydström, 2006*). Overall, these values provide a foundation for future efforts to map the metabolic state of different cell types and organelles.

## Real-time monitoring of oxidative stress

We then used NADP-Snifit to monitor changes in free $NADPH/NADP^+$ due to oxidative stress. $H_2O_2$ is a reactive oxygen species (ROS) that is metabolized into $H_2O$ and $O_2$ by different enzymes of the antioxidant system such as catalase, glutathione peroxidase and peroxiredoxin (*Veal et al., 2007*). The resulting oxidized glutathione and thioredoxin are recycled by NADPH-dependent glutathione and thioredoxin reductase, respectively. Therefore, fluctuations in $H_2O_2$ can directly influence $NADPH/NADP^+$. To observe the amplitude and kinetics of those changes, we perfused $H_2O_2$ on U2OS cells containing cytosolic NADP-Snifit. Perfusion of $H_2O_2$ produces a rapid decrease of the FRET ratio, corresponding to a decrease in $NADPH/NADP^+$ (*Figure 3*).

$H_2O_2$ itself does not influence the sensor response (*Appendix 1—figure 1p*). At the start of the experiment, free cytosolic $NADPH/NADP^+$ ratio was around 70 and incubation with 10 µM $H_2O_2$ lowers the $NADPH/NADP^+$ ratio to about 35 (*Figure 3b*). Incubation with even higher concentrations of $H_2O_2$ (100 or 200 µM $H_2O_2$) decreased the ratio further down to 20. The decrease of $NADPH/NADP^+$ reached a plateau within 5 min. The amplitude and kinetic of the $NADPH/NADP^+$ changes indicate that $H_2O_2$ scavenging by glutathione-thioredoxin antioxidant systems is a rapid and efficient process that occurs faster than regeneration of NADPH. The observation that cells even after incubation with 200 µM $H_2O_2$ maintain a cytosolic free $NADPH/NADP^+$ ratio of >10 indicates a remarkable capacity of cells to regenerate NADPH. A decrease in $NADPH/NADP^+$ ratio activates glucose-6-phosphate dehydrogenase (*Patra and Hay, 2014*) resulting in a dynamic rerouting of metabolic flux from glycolysis to the pentose phosphate pathway (*Ralser et al., 2007*; *Kuehne et al., 2015*). Remarkable is also the quick recovery of the free cytosolic $NADPH/NADP^+$ ratio after washout of $H_2O_2$. Even after incubations with 200 µM $H_2O_2$ cells return to their basal $NADPH/NADP^+$ ratio within 10 min. Finally, it is noteworthy that after washout cells initially return to a higher free $NADPH/NADP^+$ ratio than before the perfusion with $H_2O_2$ (150 versus 70) before slowly returning to the basal state (*Figure 3b*). Long-term imaging of NADP-Snifit in untreated cells showed no significant drift of the ratiometric signal for periods exceeding the time of the experiment (>2 hr), confirming the relevance of these observations. We attribute the temporarily increased $NADPH/NADP^+$ values to a metabolic adaption to oxidative stress (*Ralser et al., 2007*; *Kuehne et al., 2015*).

Our observation of oxidative stress on free $NADPH/NADP^+$ ratios is in agreement with previously reported relative changes in free $NADP^+$ (*Cameron et al., 2016*). In the same study, it was reported that incubation of cells with 100 µM $H_2O_2$ resulted in a $NADPH/NADP^+$ ratio in cell lysates of <1, as measured by a biochemical assay. In contrast, we measured free cytosolic $NADPH/NADP^+$ ratios of higher than 10 even after prolonged incubation with 100 µM $H_2O_2$, underscoring the importance of measuring NAD(P) concentrations in their biologically relevant context.

## Pharmacological alteration of cellular metabolism

Pharmacological control of cellular concentrations of nicotinamide adenine dinucleotides is of interest for numerous medical indications. For example, boosting cellular $NAD^+$ concentrations through biosynthetic $NAD^+$ precursors has been shown to increase the lifespan of multiple species and

improve numerous cellular functions (*Cantó et al., 2015*; *Verdin, 2015*). In contrast, inhibition of NAD[+] biosynthesis is pursued as a strategy to develop anticancer agents (*Kennedy et al., 2016*). However, for the large majority of such compounds, their effects on free cellular concentrations of NAD(P) remain unknown. NAD- and NADP-Snifits offer the opportunity to assess changes induced by drugs and drug candidates on free cytosolic or mitochondrial NAD[+] and NADPH/NADP[+] by flow cytometry, thus complementing the two sensors SoNar and iNAP, which permit determination of NADH/NAD[+] and NADPH through flow cytometry experiments (*Zhao et al., 2015*; *Tao et al., 2017*). We first evaluated the effect of the following NAD[+] biosynthetic precursors on free cellular NAD[+] and NADPH/NADP[+] in U2OS cells: nicotinic acid (NA), nicotinamide (NAM), nicotinamide mononucleotide (NMN) and nicotinamide riboside (NR) (*Figure 4*, *Table 2* and *Appendix 1—figure 7a*). It has been shown that the treatment of different species or cells with NAM or NR improves mitochondrial biogenesis and function (*Mouchiroud et al., 2013*; *Houtkooper et al., 2013*). In particular, NR increases total cellular and mitochondrial NAD[+] (*Cantó et al., 2012*) and moreover extends lifespan in mice (*Zhang et al., 2016*). None of the four biosynthetic precursors interacts with the sensor in vitro (*Appendix 1—figures 5*, *6*). We detected a slight but statistically significant increase in cytosolic NAD[+] in the presence of NA and NAM in U2OS cells (*Table 2*). NMN and NR increase the cytosolic free NAD[+] level to an even larger extent, as demonstrated by a 1.2- and 1.3-fold decrease in FRET ratio, respectively. The treatment of U2OS cells with NAD[+] precursors thus

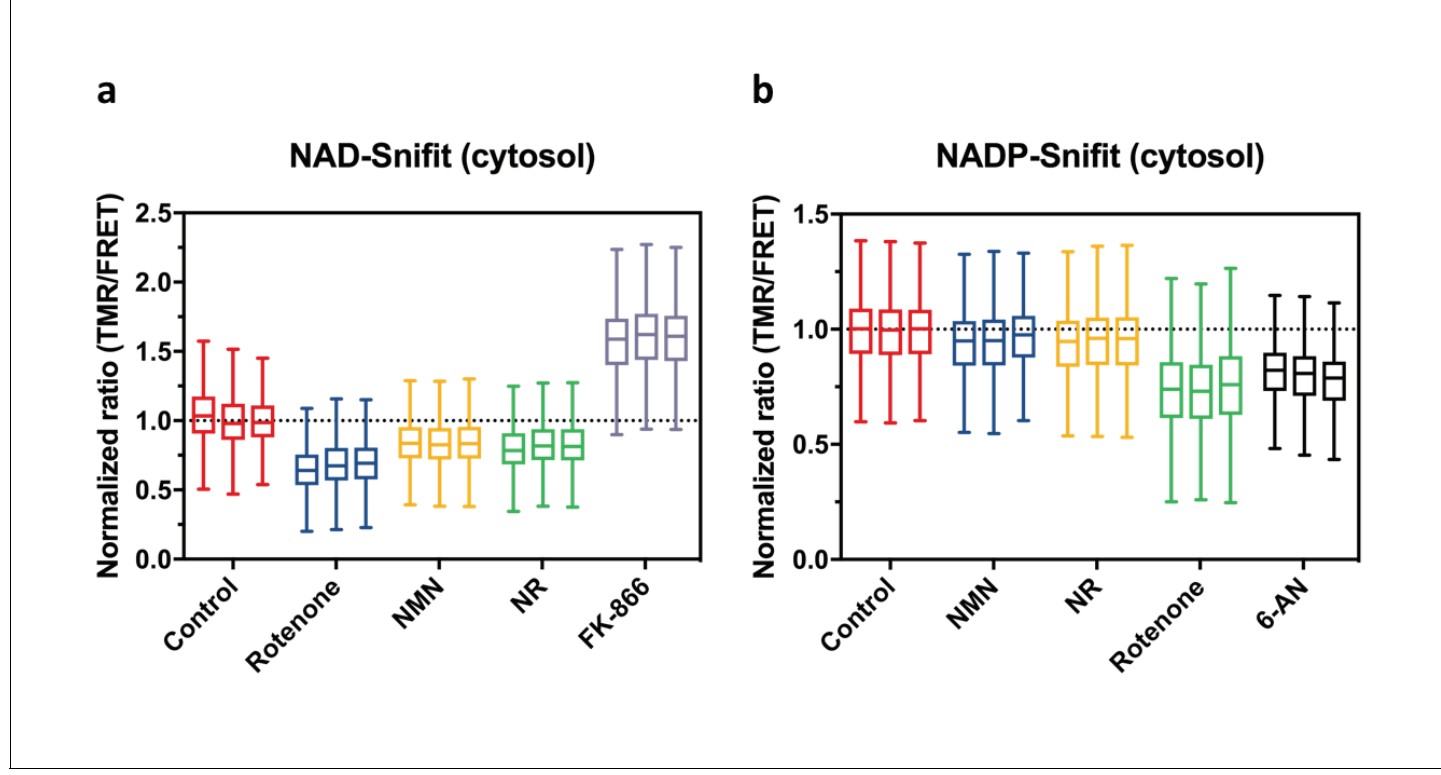

**Figure 4.** Effects of drugs and NAD biosynthetic precursors on NAD[+] and NADPH/NADP[+] levels. FRET ratios (TMR/FRET) as measured by flow cytometry of cytosolic NAD-Snifit (a) and cytosolic NADP-Snifit (b) in U2OS cells after incubation of cells under the conditions specified. For each condition, data from three independent experiments are shown to demonstrate the reproducibility of these measurements. Measured FRET ratios (TMR/FRET) are normalized to untreated control. Abbreviations and conditions: 10 µM Rotenone, 1 mM nicotinamide mononucleotide (NMN), 1 mM nicotinamide riboside (NR), 100 nM FK866, 1 mM 6-aminonicotinamide (6-AN). The Tukey-style box plots represent the 25th and 75th percentiles at the lower and upper box limits and the median as the middle bar. The whiskers extend to ± 1.5 × IQR beyond the limits of the boxes, respectively. The position of the mean is indicated by a solid square. Each data set represent n = 2000–7000 events.
DOI: https://doi.org/10.7554/eLife.32638.014

The following source data is available for figure 4:

**Source data 1.**
DOI: https://doi.org/10.7554/eLife.32638.015

has a significant effect on free $NAD^+$ leading to an estimated increase of up to 1.6-fold, but none of them show a substantial effect on $NADPH/NADP^+$ (*Table 2*). To test if free $NADPH/NADP^+$ is independent of free $NAD^+$, cells were treated with FK866. This non-competitive inhibitor of nicotinamide phosphoribosyltransferease (NAMPT) depletes free cytosolic and mitochondrial $NAD^+$, but showed no significant influence on neither cytosolic nor mitochondrial $NADPH/NADP^+$ (*Table 2*). In contrast, pharmacological inhibition of the pentose phosphate pathway with 6-aminonicotinamide (6-AN), a competitive inhibitor of glucose-6-phosphate dehydrogenase, decreases free cytosolic $NADPH/NADP^+$ (*Table 2*).

We then investigated the effect of compounds that effect cellular metabolism through other mechanisms than a direct inhibition of NAD biosynthesis. Specifically, we focused on the biguanides metformin and phenformin, an important class of anti-diabetic drugs, and two inhibitors of oxidative phosphorylation, rotenone and oligomycin A. None of these compounds interact with the sensor in vitro (*Appendix 1—figures 5*, *6*) Metformin and phenformin are known to result in activation of AMP-activated protein kinase AMPK (*Foretz et al., 2014*), which is involved in cellular energy homeostasis. Metformin is currently the most commonly used drug against type II diabetes. We measured the effect of metformin and phenformin on free $NAD^+$ and $NADPH/NADP^+$ in U2OS cells. Both compounds slightly increased cytosolic free $NAD^+$, but in contrast decreased mitochondrial $NAD^+$ levels (*Table 2*). Furthermore, incubation with metformin or phenformin reduces free cytosolic and mitochondrial $NADPH/NADP^+$ (*Table 2*). The molecular target(s) of biguanides remain unknown, but complex I of the mitochondrial electron transport chain has been proposed as one possible target (*Owen et al., 2000*). Rotenone is an established inhibitor of complex I of the mitochondrial electron transport chain. We compared its effect on free $NAD^+$ and $NADPH/NADP^+$ to those of the biguanides. As observed for the two biguanides, rotenone increased cytosolic $NAD^+$, decreased free mitochondrial $NAD^+$ and decreased both free cytosolic and mitochondrial $NADPH/NADP^+$. The similarity in the effect of biguanides and rotenone on cellular NAD(P) levels is in agreement with the proposition that biguanides inhibit complex I. The effects of inhibition of complex I on free $NAD^+$ and $NADPH/NADP^+$ are very different from those observed when inhibiting mitochondrial ATP synthase by oligomycin A, which resulted in a large mitochondrial decrease in free $NAD^+$ and large increase in free $NADPH/NADP^+$, as shown by 1.6- and 1.4-fold changes in FRET ratios, respectively (*Table 2*). These experiments demonstrate how flow cytometry measurements of free $NAD^+$ and $NADPH/NADP^+$ can be used to probe the molecular mechanisms of drugs and their effects on cellular metabolism. It should be noted that in these experiments, the properties of our sensor allowed us to reliably detect changes in FRET ratios as low as 5% (*Figure 4* and *Table 2*). This highlights their applicability in high-throughput screening approaches for compounds or genes that effect free $NAD^+$ or $NADPH/NADP^+$.

## Discussion

In this work, we introduce the first sensor for the quantification of $NADPH/NADP^+$ and a new sensor for quantifying $NAD^+$, two key biochemical parameters of cellular metabolism. The two sensors, NADP- and NAD-Snifit, consist of two synthetic fluorophores attached to self-labeling proteins and a NAD(P) binding protein, SPR. Cofactor-dependent binding of the intramolecular ligand to the SPR leads to a ratiometric FRET signal. NAD- and NADP-Snifit distinguish themselves from previously introduced 'protein-only' NAD(P) biosensors by two features: the use of synthetic fluorophores and their rational design principle. The chosen synthetic fluorophores possess excitation and emission maxima at long wavelengths, are bright and photostable, show minimal bleed-through in FRET experiments and are insensitive towards fluctuations in pH. The rational design principle of NAD(P)-Snifits permits the generation of sensors with large ratio changes and adaptable response range and colors. Together, these properties make NAD(P)-Snifits powerful tools to study the role of NAD(P) in cellular metabolism and signaling. The work also exemplifies how the synergy between synthetic chemistry and protein engineering enables the creation of hybrid molecules with unique properties. Further developments of NAD(P)-Snifits will focus on their potential use in vivo, which might require the generation of labeling substrates with increased membrane permeability.

We furthermore utilized NAD(P)-Snifits to create new insights into the biology of NAD(P). By mapping free compartmentalized $NAD^+$ levels and $NADPH/NADP^+$ ratios, we discovered that free mitochondrial $NADPH/NADP^+$ ratios are significantly higher than nucleocytoplasmic free NADPH/

NADP$^+$ ratios. Using time-lapse microscopy, we also demonstrate how single cells adapt metabolic fluxes in response to changes in environmental conditions such as oxidative stress caused by hydrogen peroxide. When exposing cells to H$_2$O$_2$-induced oxidative stress, free cytosolic NADPH/NADP$^+$ after washout of H$_2$O$_2$ initially returns to a higher NADPH/NADP$^+$ ratio than at the beginning of the experiment, indicative of increased production of NADPH through the pentose phosphate pathway (*Kuehne et al., 2015*). Finally, we demonstrate how a comparison of the relative effect of drugs on cellular NAD(P) levels can be used to test hypotheses of mechanisms of action. For example, the similar effects of the biguanides metformin and phenformin and the complex I inhibitor rotenone on compartmentalized NAD(P) pools are in agreement with the proposed inhibition of complex I by biguanides (*Owen et al., 2000*).

In summary, we introduce NAD(P)-Snifits as new, powerful tools to study the role of NAD(P) in metabolism and signaling in healthy and diseased cells.

# Materials and methods

**Key resources table**

| Reagent or resource | Source | Identifier |
| --- | --- | --- |
| Antibodies | | |
| Rabbit monoclonal anti-SPR (clone EPR9290) | Abcam | Cat#ab157194 |
| Mouse monoclonal anti-β-tubulin (clone 5H1) | BD Biosciences | Cat#556321; RRID: AB_396360 |
| Goat anti-Rabbit secondary antibody, HRP-conjugate | Cell Signaling Technology | Cat#7074; RRID: AB_2099233 |
| Horse anti-Mouse secondary antibody, HRP-conjugate | Cell Signaling Technology | Cat#7076; RRID: AB_330924 |
| Chemicals, Peptides, and Recombinant Proteins | | |
| CP-TMR-SMX | This paper | N/A |
| BG-TMR-SMX | This paper | N/A |
| SiR-Halo | This paper | N/A |
| CP-TMR | Johnsson Lab | N/A |
| Sulfapyridine (≥99%) | Sigma-Aldrich | Cat#S6252 |
| Sulfamethoxazole (>98%) | TCI | Cat#S0361 |
| Sulfachloropyridazine | Sigma-Aldrich | Cat#S9882 |
| (±)-Verapamil hydrochloride (≥99%) | Sigma-Aldrich | Cat#V4629 |
| H$_2$O$_2$ (30% (w/w), puriss. p.a.) | Sigma-Aldrich | Cat#31642 |
| 2-Deoxy-D-glucose (≥99%) | Sigma-Aldrich | Cat#D6134 |
| 6-aminonicotinamide (99%) | Sigma-Aldrich | Cat#A68203 |
| Resveratrol (>99%) | TCI | Cat#R0071 |
| Nicotinic acid (≥99.5%) | Sigma-Aldrich | Cat#72309 |
| Nicotinamide (>98%) | Sigma-Aldrich | Cat#N0636 |
| β-Nicotinamide mononucleotide (95–100%) | Sigma-Aldrich | Cat#N3501 |
| Nicotinamide riboside | Auwerx Lab, EPFL | N/A |
| FK866 hydrochloride hydrate (≥98%) | Sigma-Aldrich | Cat#F8557 |
| Metformin (97%) | Sigma-Aldrich | Cat#D150959 |
| Phenformin | Sigma-Aldrich | Cat#P7045 |
| Rotenone (≥95%) | Sigma-Aldrich | Cat#R8875 |
| Oligomycin A (≥95%) | Sigma-Aldrich | Cat#75351 |

*Continued on next page*

*Continued*

| Reagent or resource | Source | Identifier |
|---|---|---|
| NADPH tetrasodium salt (≥97%) | Roche | Cat#10621692001 |
| NADP⁺ disodium salt (≥97%) | Roche | Cat#10128058001 |
| NADH disodium salt (≥95%) | AppliChem | Cat#A1393,0001 |
| NAD⁺ free acid (100%) | Roche | Cat#10127965001 |
| ATP disodium salt (≥98%) | AppliChem | Cat#A1348,0005 |
| ADP sodium salt (≥95%) | Sigma-Aldrich | Cat#A2754 |
| GTP sodium salt hydrate (≥95%) | Sigma-Aldrich | Cat#G8877 |
| L-sepiapterin | Cayman | Cat#81650 |
| MitoTracker Green FM | Life Technologies | Cat#M7514 |
| Hoechst 33342 | Life Technologies | Cat#H1399 |
| Propidium iodide (≥94%) | Sigma-Aldrich | Cat#81845 |
| Experimental Models: Cell Lines | | |
| U-2 OS (Human osteosarcoma) | ECACC | Cat#92022711 |
| HEK-293T (Human embryonic kidney) | ATCC | Cat#CRL-3216 |
| NIH/3T3 (Mouse embroynic fibroblast) | ATCC | Cat#CRL-1658 |
| HeLa (Human cervix epitheloid carcinoma) | ATCC | Cat#CCL-2 |
| A549 (Human lung carcinoma) | ECACC | Cat#86012804 |
| Recombinant DNA | | |
| pET-51b(+) | Novagen | 71553 |
| pEBTet | (*Bach et al., 2007*) | N/A |
| pET-51b(+)_NADP | This paper | N/A |
| pET-51b(+)_NAD | This paper | N/A |
| pEBTet_NADP-cyto | This paper | N/A |
| pEBTet_NADP-nucl | This paper | N/A |
| pEBTet_NADP-mito | This paper | N/A |
| pEBTet_NAD-cyto | This paper | N/A |
| pEBTet_NAD-nucl | This paper | N/A |
| pEBTet_NAD-mito | This paper | N/A |
| Software and Algorithms | | |
| OriginPro 9 | OriginLab Corporation | http://www.originlab.com/ |
| PyMOL | Schrödinger, LLC | https://www.pymol.org/ |
| FIJI (ImageJ) | (*Schindelin et al., 2012*) | https://fiji.sc/ |
| SymPhoTime 64 | PicoQuant | https://www.picoquant.com/ |
| Huygens Essential | Scientific Volume Imaging | https://svi.nl/HuygensEssential |
| FlowJo v10 | FlowJo, LLC | https://www.flowjo.com/ |
| R 3.4.0 | R Core Team, 2017 | https://www.r-project.org/ |
| Other | | |
| Leica TCS SP8 X confocal microscope - PicoHarp 300 (PicoQuant) TCSPC module | Leica/PicoQuant | http://www.leica-microsystems.com https://www.picoquant.com/ |
| IN Cell Analyzer 2200 automated widefield microscope | GE Healthcare Life Sciences | http://www.gelifesciences.com/ |
| Leica DMI6000B widefield microscope | Leica | http://www.leica-microsystems.com |

## Chemical synthesis and sensor constructs

Detailed procedures for the synthesis of the SNAP-tag substrates and plasmids construction can be found in the Appendix 1 Information. Synthesis of SiR-Halo has been described previously (*Lukinavičius et al., 2013*).

## Bacterial protein expression, purification and labeling

The sensor proteins were expressed in transformed *Escherichia coli* strain Rosetta-gami 2(DE3) (Novagen). Bacterial cultures were grown in selective (100 µg/mL ampicillin) LB medium at 37 °C to an $OD_{600nm}$ of 0.8, cooled down to 16°C prior to induction with 1 mM isopropyl β-D-thiogalactopyranoside (IPTG). After 16 hr, the cells were harvested by centrifugation, lysed by sonication in presence of a protease inhibitor cocktail (cOmplete-EDTA-free, Roche) and the resulting cell lysates were cleared by centrifugation. The proteins were purified by two successive purification steps using Ni-NTA (Qiagen) and Strep-Tactin (IBA) columns according to the supplier's instructions. The purified proteins can be stored for several months at a concentration of 50–100 µM at −80 °C as flash frozen ($N_2$ liq.) small aliquots (50 µL) prepared in 50 mM HEPES, 150 mM NaCl, 1 mM DTT, 5% (v/v) glycerol, pH 7.5 or at −20 °C as stocks prepared in 50 mM HEPES, 150 mM NaCl, 1 mM DTT, 50% (v/v) glycerol, pH 7.5. For sensor labeling, the sensor protein was diluted to 5 µM in buffer (50 mM HEPES, 150 mM NaCl, pH 7.5) with 10 µM BG-TMR-SMX and 10 µM SiR-Halo and incubated at room temperature for 1 hr. The excess of SNAP-tag and Halo-tag substrates were removed by gel filtration using NAP-5 Sephadex prepacked columns (GE Healthcare). The final concentration of labeled sensor proteins was determined by measuring the absorbance at 555 nm and 650 nm in the labeling buffer supplemented with 0.1% SDS ($\varepsilon(TMR)_{555nm}$ = 90,000 $M^{-1}cm^{-1}$, $\varepsilon(SiR)_{650nm}$ = 100,000 $M^{-1}cm^{-1}$).

## Titrations of the sensors

The labeled sensors were diluted to a concentration of 20 nM in 100 µL of buffer (unless specified 50 mM HEPES, 150 mM NaCl, 0.5 mg/mL BSA, pH 7.5) containing defined concentrations of analytes ($NADP^+$, $NAD^+$ or $NADPH/NADP^+$) in black non-binding 96-well plates (Greiner Bio-One). The solutions were incubated at room temperature for at least 15 min to ensure that the sensor conformation had reached equilibrium. Fluorescence measurements were performed on an Infinite M1000 spectrofluorometer (TECAN). Both the excitation and emission bandwidth for all measurements were set to 10 nm. For the sensor constructs labeled with TMR and SiR, the emission spectra were recorded from 540 nm to 740 nm using a step size of 1 nm with an excitation at 520 nm. For the sensor constructs with EGFP and TMR, the emission spectra were measured from 480 nm to 610 nm using a step size of 1 nm with an excitation of 450 nm. The emission ratios of the FRET donor over FRET acceptor (TMR/SiR: 577 nm/667 nm; EGFP/TMR: 508 nm/577 nm) were measured as technical triplicates and were plotted as mean ± s.d. against the analyte concentration. The plots were fitted using a single binding isotherm (*Equation 3*) to obtain the $c_{50}$ and the maximum FRET ratio change ($\Delta R_{max}$ = $R_{max}/R_{min}$). The $c_{50}$ values and maximum ratio changes are reported as mean ± s.d. from three independent titrations.

$$R = R_{max} + \frac{R_{min} - R_{max}}{1 + \frac{c_{50}}{[Analyte]}} \tag{3}$$

$$R = R_{min} + \frac{R_{max} - R_{min}}{1 + \frac{r_{50}}{[Analyte]}} \tag{4}$$

with R being the experimental emission ratio of donor vs acceptor, [Analyte] the concentration of cofactors, $R_{max}$ and $R_{min}$ are the maximum and minimum emission ratio corresponding to the open (free) and closed (saturated) sensor, respectively. Fits were performed using OriginPro 2017 (OriginLab Corporation) with $R_{max}$, $R_{min}$, $c_{50}$ as free parameters.

For the titrations using $NADPH/NADP^+$, the total cofactor concentration was fixed to 100 µM while varying the ratios of NADPH vs $NADP^+$ and the plots were fitted using the single binding isotherm (*Equation 4*) to determine $r_{50}$ defined as the $NADPH/NADP^+$ ratio corresponding to half-maximal sensor response. The prepared ratios $NADPH/NADP^+$ were corrected by measuring the

percentage of NADP$^+$ present in the commercial stock of NADPH (NADPH-RO, Roche) by absorbance as described in the Supplementary Note 2. To obtain higher NADPH/NADP$^+$ ratios, the NADPH was purified by anion-exchange chromatography using a Resource Q column (GE Healthcare) and freshly used for titrations. The plots were fitted by fixing $R_{max}$ determined by addition of a saturating concentration of competitive free ligand (2 mM sulfamethoxazole), while setting the other parameters free.

It has to be noted that the FRET donor and acceptor possess different dynamic ranges, therefore their respective emission ratio is not linearly correlated with the sensor occupancy as described previously (*Pomorski et al., 2013*). The determination of the sensor's $K_D'$ was performed by normalizing the individual fluorescence intensities of TMR or SiR by the sensor's isosbestic point (645 nm) and fitted with the previously described *Equations (3)* or (4), where $c_{50}$, $r_{50}$ are replaced by $K_D'$ or $K_{50}$. $K_{50}$ is defined as the NADPH/NADP$^+$ ratios corresponding to sensor's half-saturation with NADP$^+$.

## Cell culture, transfection and cell labeling

U2OS, HEK293T, NIH/3T3, HeLa cells were cultured in high-glucose DMEM with GlutaMAX-I, 1 mM pyruvate (Gibco) supplemented with 10% HyClone FetalClone II Serum (GE Healthcare) at 37 °C in a humidified incubator at 5% $CO_2$. Cells were subcultured twice per week or at 90% confluency using StemPro Accutase (Gibco, Life Technologies). The cells are not known to be misidentified no cross-contaminated. The cell lines are regularly checked and not infected with mycoplasma.

To generate semi-stable cell lines, the cells were transfected with the pEBTet expression vectors using Lipofectamine 3000 according to the manufacturer's instruction. 48 hr after transfection, the cells were selected with the full growth medium supplemented with 1 μg/mL puromycin for one week. After the selection, the amplified transfected cells were continuously maintained in selective conditions and stocks were frozen in 10% DMSO at low passage numbers and stored at −80 °C for further use. Cell lines were regularly checked for mycoplasma infection (biochemical test: MycoAlert, Lonza and imaging: Hoechst 33342 staining at 0.1 μg/mL) and used for experiments before 25 passages. Expression of the sensor proteins were induced with 100 ng/mL doxycycline for the cytosolic and nuclear sensors and 10 ng/mL doxycycline for the mitochondrial localized sensor for 24 hr, after which the cells were labelled with 1 μM fluorescent substrates (CP-TMR-SMX, SiR-Halo) in fresh prewarmed full growth medium supplemented with 10 μM (±)-verapamil hydrochloride (Sigma-Aldrich) overnight at 37 °C, 5% $CO_2$. Then, the excess of dyes was removed by washing cells three times with full growth medium followed by 2 hr incubation. The medium was exchanged one last time before imaging. The fluorescent substrates (CP-TMR-SMX, SiR-Halo) are prepared as 2 mM DMSO stock (2000x). (±)-verapamil is prepared as 10 mM stock (1000x) in cell culture grade water and sterile filtered.

## Live-cell quantification of NADPH/NADP$^+$ and NAD$^+$ by ratio imaging

Semi-stable U2OS cell lines (NADP-Snifit: cytosol, nucleus and mitochondria and NAD-Snifit: cytosol) were passaged with StemPro Accutase (Gibco, Life Technologies) and plated ($10^4$ cells/well) in poly-D-Lysine coated glass-bottom 96-well plates (MatTek Corporation) and cultured in full growth medium at 37 °C, 5% $CO_2$. The next day, the expression of the different constructs were induced with 100 ng/mL doxycycline for the cytosolic, nuclear sensors and 10 ng/mL doxycycline for the mitochondrial sensor. After 24 hr, the sensor proteins were labeled with 1 μM CP-TMR-SMX, 1 μM SiR-Halo and 10 μM (±)-verapamil overnight (16 hr). The excess of labeling compounds were washed three times with phenol red free full growth medium and the cells were incubated 2 hr at 37 °C, 5% $CO_2$ before imaging. The cells were imaged before and after being treated with 2 mM sulfapyridine (use to fully open the sensors in situ) on a IN Cell Analyzer 2200 (GE Healthcare) widefield automated microscope equipped with a sCMOS camera (2048 × 2048 pixels) using either Nikon Plan Apo 20X/0.75 CFI/60 or Plan Fluor 40X/0.60 CFI/60 air-objectives and three channels per image acquisition: Cy3/Cy3 (TMR channel), Cy3/Cy5 (FRET channel) and Cy5/Cy5 (SiR channel), with filters specification: Cy3: excitation (542/27 nm), emission (597/45 nm); Cy5: excitation (632/22 nm), emission (684/25 nm) using 200 ms exposure time at 37 °C, 5% $CO_2$. Image analyses were performed in FIJI (*Schindelin et al., 2012*). Fluorescence images in each channel were first flat-field (using flat-field reference images) and background (by subtracting the fluorescence intensity of ROIs corresponding to background region) corrected. Then, FRET images were corrected for bleed-through according

to the previously determined (*Spiering et al., 2013*) *Equation (5)* using single-labeled controls to determine the donor emission ratio α (i.e. bleed-through of the donor into the acceptor channel using a donor-only sample) and β (i.e. direct acceptor excitation from TMR excitation light using an acceptor-only sample). Due to the large spectral separation between the FRET pairs, α and β are very small correction coefficients. α and β were determined to be 0.054 and 0.051 with this microscopy setup.

$$FRET_c = FRET_{raw} - \alpha \cdot TMR - \beta \cdot SiR \tag{5}$$

The emission ratios (TMR/FRET$_c$) of 60 individual cells from three different cell preparations were tracked and measured before and 15 min after the treatment of 2 mM sulfapyridine. Sulfapyridine (SPY) treatment allows to fully open the sensors in situ and to determine the normalized FRET ratio change ΔR (ΔR = R$_{SPY}$/R$_{basal}$). ΔR values were used to convert the emission ratio corresponding to the apparent sensor occupancy R of the cells at basal state (R = R$_{max}$/ΔR) as the dynamic range of the sensor of the instrumental setup is similar to in vitro measurements. NADPH/NADP$^+$ ratios and NAD$^+$ are quantified using the following *Equations (6 and 7)*, where R$_{max}$, R$_{min}$, r$_{50}$ and c$_{50}$ are parameters determined by in vitro titrations at 37 °C (NADP-Snifit: R$_{max}$ = 4.58 ± 0.12, R$_{min}$ = 0.52 ± 0.02, r$_{50}$ = 30 ± 3; NAD-Snifit: R$_{max}$ = 4.47 ± 0.16, R$_{min}$ = 0.59 ± 0.03, c$_{50}$ = 130 ± 14 μM).

$$\frac{[NADPH]}{[NADP^+]} = r_{50}\frac{R - R_{min}}{R_{max} - R} \tag{6}$$

$$[NAD^+] = c_{50}\frac{R_{max} - R}{R - R_{min}} \tag{7}$$

## Live-cell quantification of NADPH/NADP$^+$ and NAD$^+$ by FLIM

Semi-stable U2OS cell lines for sensors expression in the different subcellular compartments (cytosol, nucleus, mitochondria) were passaged with StemPro Accutase (Gibco, Life Technologies), plated in poly-D-Lysinecoated glass-bottom 12-well plates (MatTek Corporation) and cultured in full growth medium at 37 °C, 5% CO$_2$. Sensors expression were induced with 10 (mitochondria targeted sensors) or 100 ng/mL doxycycline. After 24 hr, the sensor constructs were labeled overnight (16 hr) either only with 1 μM CP-TMR-SMX (for donor only controls) or with 1 μM CP-TMR-SMX and 1 μM SiR-Halo each time in presence of 10 μM (±)-verapamil. The cells were washed three times in full growth medium, incubated for another 2 hr before imaging. Fluorescence lifetimes measurements were performed on a laser scanning confocal microscope (Leica TCS SP8 X) equipped with an 63x oil-immersion objective (HC PL APO 63x/1.40 CS2) and a PicoHarp 300 (PicoQuant) TCSPC module. As excitation source, the white-light laser was set 514 nm with 20 MHz pulse frequency. The FRET donor emission was measured on a hybrid photodetector for single molecule detection (Leica HyD SMD) with a detection range of 550–610 nm. The images were typically acquired using 180 × 180 μm (cytosol, nucleus) or 70 × 70 μm (mitochondria) with 512 × 512 pixels, scan speed 100 Hz, pinhole at one airy unit, a laser power adjusted to 10$^5$ average photon counts per second to avoid pile-up effects and a target photon counts of 500/pixel. All the measurements were performed at 37 ± 1 °C. The data acquisition and analysis were performed using SymPhoTime 64 (PicoQuant). The fluorescence decays of individual cells were extracted by ROIs (sum of the photons of all the pixels of a ROI, typically with 10$^6$ photon counts) and were fitted using an n-exponential reconvolution model (*Equation 8*);

$$y(t) = \sum_{i=0}^{n-1} IRF \otimes \bigg|_{Bkgr_{IRF}|Shift_{IRF}} \alpha_i \, exp\left(\frac{-t}{\tau_i}\right) + Bkgr_{Dec} \tag{8}$$

where the instrument response function (IRF) was calculated from the convolution integral of the model function. Bkgr$_{IRF}$, Shift$_{IRF}$, Bkgr$_{Dec}$ correspond to the corrections for the IRF background and displacement and decay background. α$_i$ and τ$_i$ correspond to the pre-exponential factors and the lifetimes. The goodness-of-fit was determined by the reduced chi-square (χ$^2$ < 1.2) using a nonlinear least-squares analysis and examining the weighted residuals trace. The donor-only and FRET samples were fitted according to a bi-exponential and third-order exponential fitting model. An example of

fluorescence decays and fitting can be found in *Appendix 1—figure 4*. The amplitude weighted average lifetimes <τ> (*Equation 9*) were used to calculate the FRET efficiencies (*Equation 10*) before (E, at basal cellular state) and after the treatment of cells with 2 mM sulfapyridine representing the minimal FRET efficiency ($E_{min}$).

$$\langle \tau \rangle = \frac{\sum \alpha_i \tau_i}{\alpha_i} \tag{9}$$

$$E = 1 - \frac{\langle \tau_{DA} \rangle}{\langle \tau_D \rangle} \tag{10}$$

<τ$_{DA}$> and <τ$_D$> represent the amplitude weighted average lifetimes for the FRET and donor-only samples. The lifetimes measured in vitro and in U2OS cells and reported in *Appendix 1—tables 5* and *6*, respectively, represent the mean ± s.d. of 10 individual cells from three independent experiments (n = 10). NADPH/NADP$^+$ ratios and NAD$^+$ are quantified using *Equations (1 and 2)*, where E and $E_{min}$ correspond to the FRET efficiency of the sensor in situ prior (basal state) and after the treatment with 2 mM sulfapyridine and $E_{max}$ was determined with the same setup using the purified sensor with saturating concentration of cofactor. $K_{50}$ and $K_D'$ are the NADPH/NADP$^+$ ratio and NAD$^+$ concentration corresponding to sensor's half-saturation determined from in vitro titrations at 37 °C (NADP-Snifit: $K_{50}$ = 11.6 ± 3.3, NAD-Snifit: $K_D'$=363 ± 47 µM).

## Real-time monitoring of oxidative stress

Semi-stable U2OS cells (cytosolic NADP-Snifit) were plated on poly-L-ornithinecoated glass coverslips (VWR 20 × 20 mm) using a 6-well plate and cultured in full growth medium at 37 °C, 5% CO$_2$. Sensor expression was induced the next day by addition of 100 ng/mL doxycycline. After 24 hr, the protein construct was labeled with 1 µM CP-TMR-SMX, 1 µM SiR-Halo and 10 µM (±)-verapamil in full growth medium overnight (16 hr). The cells were washed three times with full growth medium and incubated 2 hr at 37 °C, 5% CO$_2$. The medium was exchanged for HBSS (Lonza) 30 min before imaging. Glass coverslips were transferred to a Cytoo chamber (44 × 34×10 mm). Time-course experiments of sensor imaging were performed on a Leica DMI6000B wide-field microscope equipped with a Hamamatsu-C9100 EM-CCD camera and a 40x oil-immersion objective (HCX PL APO 40.0 × 1.25). Gravity fed perfusion of the chamber was performed at a flow rate of 1 mL/min. For each frame, the two channels (donor and FRET) were measured consecutively, with an interval of 10 s between individual frames. Cy3 was used as excitation filter (530/35 nm) and the emission filters were respectively Cy3 (580/40 nm) for the donor channel and Cy5 (700/72 nm) for the acceptor channel. The perfused solutions (A = 2 mM sulfapyridine, B = 10 µM H$_2$O$_2$, C. 100 µM H$_2$O$_2$, D. 200 µM H$_2$O$_2$) were all prepared in HBSS (Lonza). HBSS solution was continuously perfused during the other point of the experiment. For image analysis, the 16-bit images (306 × 306 µm, 512 × 512 pixels) were background corrected and fluorescence intensity time-traces from 10 cells (defined as ROIs) were extracted for the TMR and FRET channels using FIJI (*Schindelin et al., 2012*). For each cells and time points, the ratio (TMR/FRET) was calculated. A graph of the emission ratio (TMR/FRET) vs. time was generated as mean ± s.d (n = 10 cells).

## Flow cytometry measurements

10$^4$ semi-stable U2OS cells (NAD- and NADP-Snifit: cytosol, mitochondria) were plated in 96-well culture plates (TTP U-bottom plates) using 200 µL DMEM high glucose (GlutaMax-I, 10% FetalClone II, 1 mM sodium pyruvate) supplemented with 10 (for mitochondrial sensors) or 100 ng/mL doxycycline (for cytosolic sensors) to induce proteins expression. The constructs were labeled with 1 µM CP-TMR-SMX, 1 µM SiR-Halo and 10 µM (±)-verapamil in full growth medium overnight (16 hr). After exchanging three times the medium to remove the excess of dyes, the cells were treated for 24 hr in different conditions. The different compound were prepared in DMEM high glucose (GlutaMax-I, 10% FetalClone II, 1 mM sodium pyruvate). Then, the cells were washed with PBS and detached with 20 µL StemPro Accutase (Gibco, Life Technologies) for 5 min at 37 °C. The cells were resuspended and separated by gentle mixing with a multichannel pipette using 120 µL growth medium (in treatment condition) and 10,000 cells were analyzed on a LSR II flow cytometer (BD Biosciences) equipped with HTS module. The different lasers and filters were used to record the donor, FRET and

acceptor fluorescence: 561 nm laser with 585/15 nm filter for TMR, 561 nm laser with 660/20 nm filter for FRET and 640 laser with 670/20 nm filter for SiR. Unstained cells and induced cells only labeled with either the donor or acceptor dye were used to measure fluorescence spillover. Sensor labeled with CP-TMR and SiR-Halo (forming essentially a non-functional sensor) was used as additional control to test eventual nonspecific ratio change due to the added compounds (e.g. quenching, increased fluorescence). The cell viability for the different treatment was tested by propidium iodide staining. The data were analyzed on FlowJo software. Gating strategy involved the removal of dead cells and debris (SSC-A vs FCS-A), doublets removal (SSC-A vs SSC-W) and selection of the labeled cell population (SiR vs TMR). The gated cells population in the different conditions were analyzed by determining the median of their TMR/FRET ratio. For each condition, the median was averaged from three measurements obtained from different cell preparation. The final results are represented as mean TMR/FRET ratios $\pm$ s.d from three independent experiments. For each condition, the mean ratios were normalized with the untreated cells. An example of the gating strategy and the distribution of TMR/FRET ratio of cell populations using different treatment can be found in *Appendix 1—figure 7a*. As we cannot experimentally determine $R_{min}$, $c_{50}$ and $r_{50}$ values of our sensors on the flow cytometer and would have to use the parameters determined on a different instrument to transform FRET ratios in concentrations or ratios (*Appendix 1—table 4*), concentrations or ratios obtained this way should only be considered as estimates.

### Quantification and statistical analysis

Titrations data (*Figure 2* and *Appendix 1—figure 1*) are represented as mean $\pm$ s.d. of the emission ratio (TMR/SiR) from technical triplicates. The calculated fitting parameters ($c_{50}$, $r_{50}$, $K_D'$, $K_{50}$, $R_{min}$, $R_{max}$) used for the quantification of $NAD^+$ and $NADPH/NADP^+$ by ratio imaging, FLIM and flow cytometry (estimations) were determined as mean $\pm$ s.d. of three independent titrations (each performed in triplicates) (*Table 1*). Flow cytometry data (*Figure 4* and *Appendix 1—figure 7*) were characterized by non-normal distributions. In essence, the sample distributions showed a positive kurtosis and skewness, and were heteroscedastic. The statistical analysis (*Appendix 1—figure 7*) was then performed in R by a Kruskal-Wallis test with post-hoc Dunn's test using the Benjamini-Hochberg method (FDR) for multiple comparison correction with respect to control conditions. The significance level was set to $\alpha = 0.05$ and two-tailed p-values were reported (* $p < 0.05$; n.s. $p \geq 0.05$).

## Acknowledgements

The authors acknowledge support of the Ecole Polytechnique Fédérale de Lausanne, the Swiss National Science Foundation, and the Max-Planck Society. We are grateful to Johan Auwerx for providing us with a sample of nicotinamide riboside. We thank Luigi Bozzo (EPFL) and Loïc Tauzin (EPFL) for technical assistance. We are grateful to all members of the Johnsson lab for critical reading of the manuscript.

## Additional information

### Competing interests

Olivier Sallin, Luc Reymond, Kai Johnsson: has filed a patent application (WO2016131833A1) on the design and use of sensors for the detection of NAD(P). The other authors declare that no competing interests exist.

### Funding

| Funder | Grant reference number | Author |
|---|---|---|
| Max-Planck-Gesellschaft | Institutional support and open-access funding | Kai Johnsson |
| École Polytechnique Fédérale de Lausanne | Institutional support and open-access funding | Corentin Gondrand |

The funders had no role in study design, data collection and interpretation, or the decision to submit the work for publication.

## Author contributions
Olivier Sallin, Conceptualization, Investigation, Writing—original draft, Writing—review and editing; Luc Reymond, Conceptualization, Investigation, Writing—review and editing; Corentin Gondrand, Fabio Raith, Birgit Koch, Investigation, Writing—review and editing; Kai Johnsson, Conceptualization, Writing—original draft, Writing—review and editing

## Author ORCIDs
Fabio Raith http://orcid.org/0000-0002-7235-8453
Kai Johnsson http://orcid.org/0000-0002-8002-1981

## Decision letter and Author response
Decision letter https://doi.org/10.7554/eLife.32638.045
Author response https://doi.org/10.7554/eLife.32638.046

## Additional files

### Supplementary files
• Transparent reporting form
DOI: https://doi.org/10.7554/eLife.32638.016

### Data availability
All data generated or analysed during this study are included in the manuscript and supporting files. Source data files have been provided for Figures 2, 3, & 4, and Appendix 1-Figures 1, 3, & 7.

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

# Appendix 1

DOI: https://doi.org/10.7554/eLife.32638.017

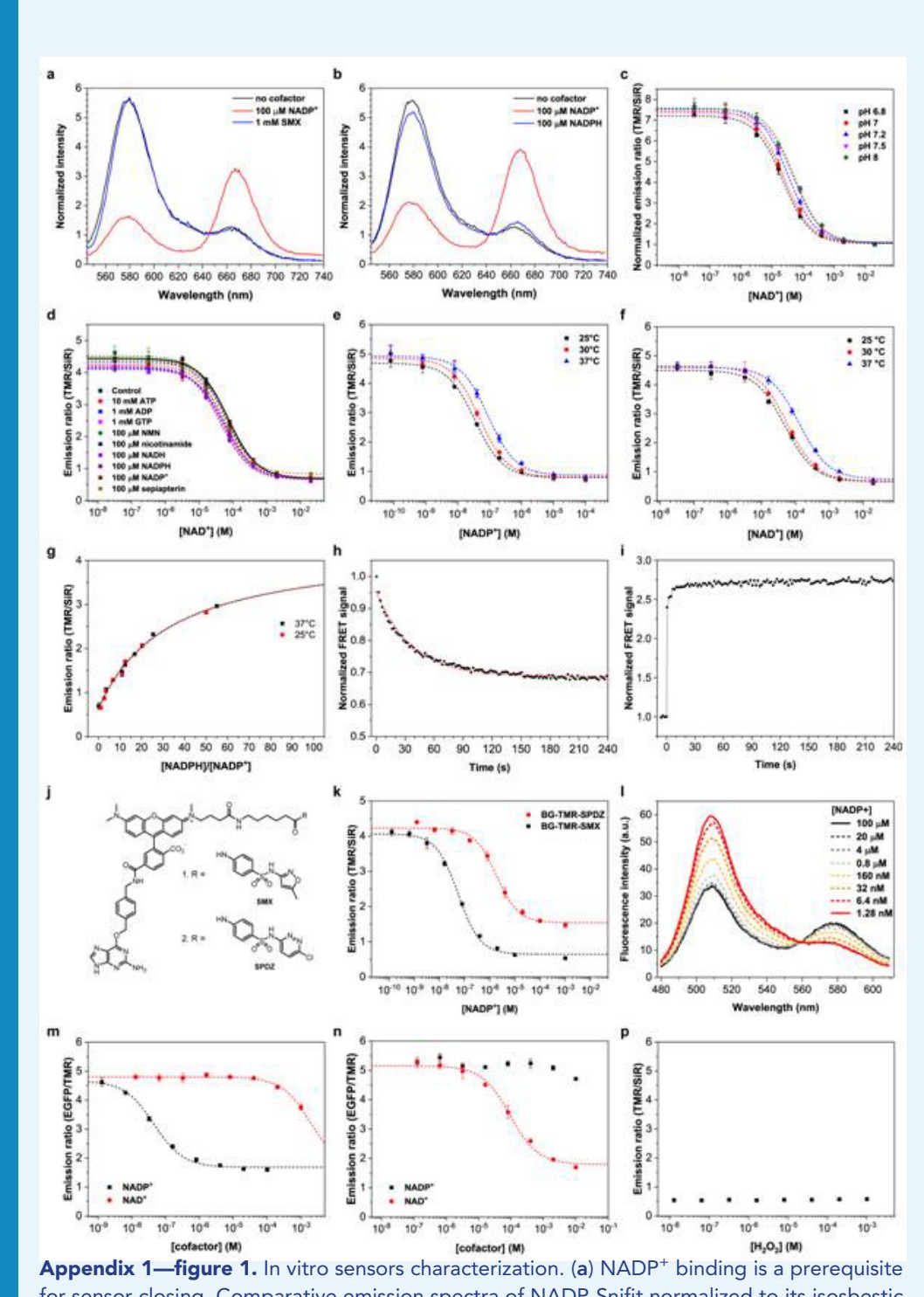

**Appendix 1—figure 1.** In vitro sensors characterization. (**a**) NADP$^+$ binding is a prerequisite for sensor closing. Comparative emission spectra of NADP-Snifit normalized to its isosbestic point (645 nm) in absence of NADP$^+$ (black line), in presence of 100 μM NADP$^+$ (red line) and in presence of 100 μM NADP$^+$ and 1 mM sulfamethoxazole (SMX) (blue line). (**b**) The intramolecular ligand does not bind to the sensor saturated with NADPH. Emission spectra of NADP-Snifit without NADP$^+$ (black line), after the addition of 1 mM glucose-6-phosphate and 100 μM NADP$^+$ (red line) and finally after a 30 min incubation in presence of 1 nM glucose-6-

phosphate dehydrogenase (G6PD). The conversion of $NADP^+$ into NADPH is not fully complete as G6PD is inhibited at high $NADPH/NADP^+$ ratios. However, obtaining pure NADPH is difficult since most commercial stock of NADPH were found to have ~2–3% $NADP^+$ as impurity. (**c**) Titrations of NAD-Snifit with $NAD^+$ at various pH ranging from 6.8 to 8.0. (**d**) Titrations of NAD-Snifit with $NAD^+$ in presence of a fixed concentration of one of the listed different metabolites and structurally close molecules and the substrate sepiapterin. (**e**) Titrations of NADP-Snifit with $NADP^+$ at 25°C, 30°C and 37°C ($c_{50}$ varies from 35 ± 3 nM to 88 ± 7 nM, from 25°C to 37°C) (**f**) Titrations of NAD-Snifit with $NAD^+$ at 25°C, 30°C and 37°C ($c_{50}$ varies from 63 ± 12 µM to 130 ± 14 µM, from 25°C to 37°C). (**g**) Titrations of NADP-Snifit with varying $NADPH/NADP^+$ ratios at 25°C and 37°C. The $r_{50}$ of the fitted curves do not change significantly between the two temperatures ($r_{50}$ is 32 and 33, respectively for 25°C and 37°C). (**h**) Kinetics of sensor opening. The experiment is conducted by injection of 5 mM NADPH at time zero to the closed sensor saturated with $NADP^+$ (100 nM sensor, 10 µM $NADP^+$). The measured $t_{1/2}$ fitted with a single-exponential decay is 25 s. (**i**) Time course of the sensor closing following the injection of 1 mM $NADP^+$ at the zero time point. The experimental set-up does not resolve the closing kinetic for the unsaturated sensor. (**j**) Chemical structures of BG-TMR-SMX (1) and BG-TMR-SPDZ (2). (**k**) Titrations of NADP-Snifit labeled either with BG-TMR-SMX or BG-TMR-SPDZ with $NADP^+$. The determined $c_{50}$ values of the sensor for $NADP^+$ are of 29 ± 7 nM for sulfamethoxazole (SMX) and 1.9 ± 0.3 µM for sulfachloropyridazine (SPDZ) as intramolecular ligand. (**l**) Emission spectra of the EGFP sensor version SPR(WT)-EGFP-p30-SNAP titrated with $NADP^+$. (**m**) Titration of SPR-EGFP-p30-SNAP with $NADP^+$ and $NAD^+$. Similarly to NADP-Snifit, the fitted $c_{50}$ is of 45 nM and ~2 mM (extrapolated), respectively for $NADP^+$ and $NAD^+$. (**n**) Titration of SPR(D41W42)-EGFP-p30-SNAP with $NADP^+$ and $NAD^+$. The sensor is specific for $NAD^+$ with a fitted $c_{50}$ of 63 ± 12 µM. (**p**) NADP-Snifit was titrated up to 1 mM $H_2O_2$ with a fixed concentration of $NADP^+$. Unless indicated, the measurements were performed in 50 mM HEPES, 150 mM NaCl, 0.5 mg/mL BSA, pH 7.4 at 25°C. Data represent the mean ± s.d. of titrations performed in triplicate.

DOI: https://doi.org/10.7554/eLife.32638.018

The following source data is available for figure :

**Appendix 1—figure 1—source data 1**
DOI: https://doi.org/10.7554/eLife.32638.019
**Appendix 1—figure 1—source data 2.**
DOI: https://doi.org/10.7554/eLife.32638.020
**Appendix 1—figure 1—source data 3.**
DOI: https://doi.org/10.7554/eLife.32638.021
**Appendix 1—figure 1—source data 4.**
DOI: https://doi.org/10.7554/eLife.32638.022
**Appendix 1—figure 1—source data 5.**
DOI: https://doi.org/10.7554/eLife.32638.023
**Appendix 1—figure 1—source data 6.**
DOI: https://doi.org/10.7554/eLife.32638.024
**Appendix 1—figure 1—source data 7.**
DOI: https://doi.org/10.7554/eLife.32638.025
**Appendix 1—figure 1—source data 8.**
DOI: https://doi.org/10.7554/eLife.32638.026
**Appendix 1—figure 1—source data 9.**
DOI: https://doi.org/10.7554/eLife.32638.027
**Appendix 1—figure 1—source data 10.**
DOI: https://doi.org/10.7554/eLife.32638.028

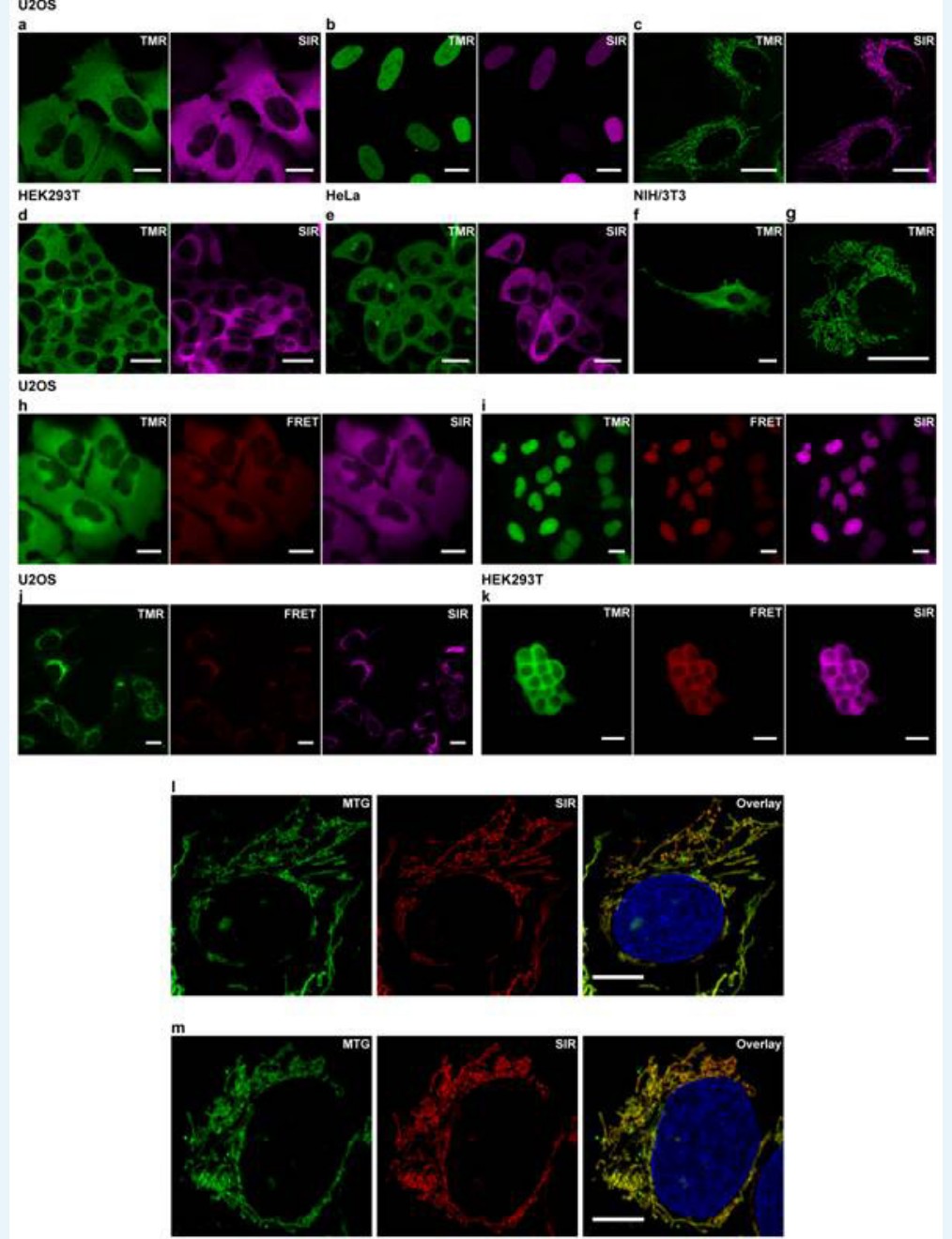

**Appendix 1—figure 2.** Live-cell imaging with NAD(P)-Snifit. Multichannel fluorescence confocal images of NAD(P)-Snifit localized in the cytosol (**a**), nuclei (**b**) and mitochondria (**c**) of U2OS cells. Confocal images of cytosolic NAD(P)-Snifit in HEK293T (**d**), HeLa (**e**) and NIH/3T3 (**f**) cells. (**g**) Confocal images of NADP-Snifit localized in the mitochondria of NIH/3T3 cells. Widefield images of NAD(P)-Snifit localized in the cytosol (**h**), nuclei (**i**) and the mitochondria (**j**) of U2OS cells and in the cytosol of HEK293T cells (**k**). All images were taken in full growth medium (DMEM +10% FBS). The different detection channels are represented using pseudocolors: donor channel (TMR, green), FRET channel (red) and the emission of acceptor through direct excitation (SiR, magenta). All images were subject to background correction and the FRET channel was additionally corrected for crosstalk. Scale bars, 20 µm. (**l–m**) Colocalization of mitochondrial NADP-Snifit with MitoTracker Green. Three color confocal images of MitoTracker Green (MTG, green), mitochondrial localized NADP-Snifit (red) using the acceptor dye channel (SiR) and Hoechst 33342 as nuclear stain (blue) in living U2OS cells.

The images were deconvolved using Huygens Essentials package prior to the colocalization analysis. The Pearson's coefficient between the MTG and SiR channels are 0.80 (**l**) and 0.90 (**m**). Scale bars, 10 μm.

DOI: https://doi.org/10.7554/eLife.32638.029

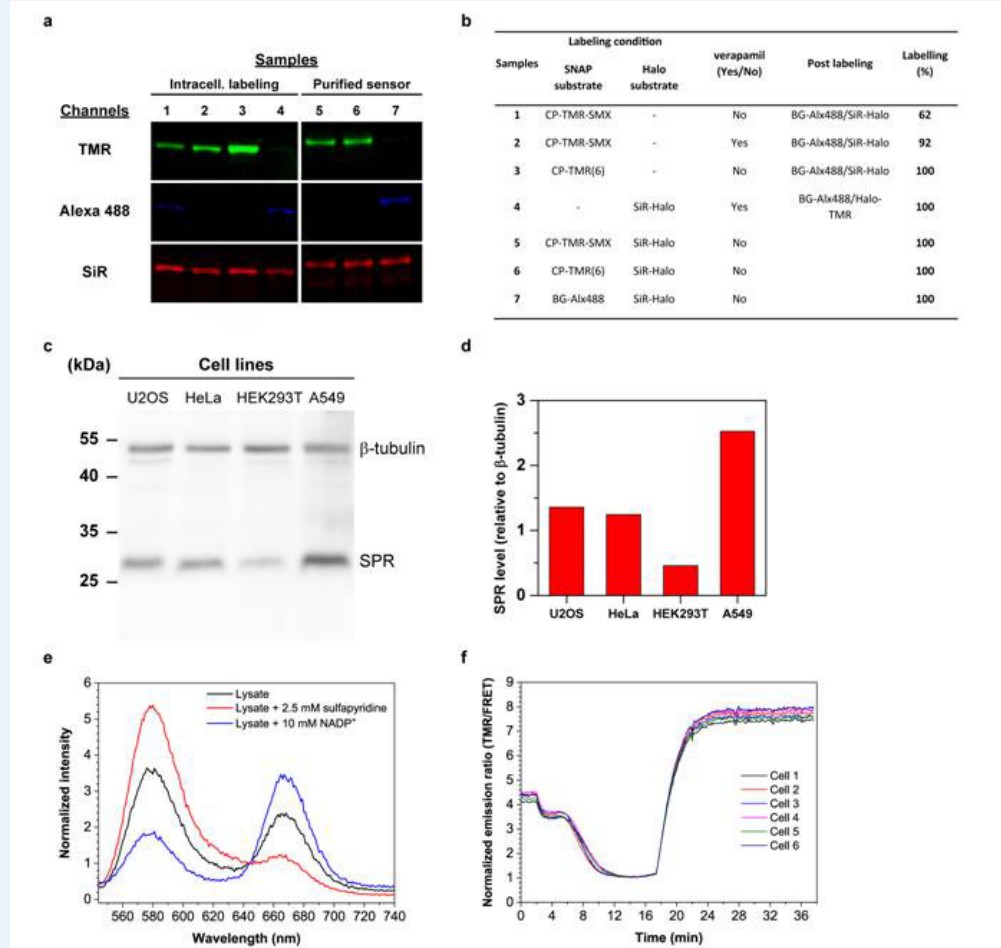

**Appendix 1—figure 3.** In cellulo sensor characterization. Intracellular labeling efficiencies. (**a**) Representative in-gel fluorescence detection of intracellular and in vitro (control) sensor protein labeling. The sensor protein is labeled intracellularly (U2OS cells) with CP-TMR-SMX, CP-TMR(6) or SiR-Halo (samples 1–4) with or without the presence of the efflux pump inhibitor verapamil (10 μM), overnight. The cells were washed and lysed with an excess BG-Alexa(488) and SiR-Halo or Halo-TMR to quantify the unlabeled fraction of SNAP-tag and Halo-tag. As control the purified sensor was labeled in vitro with CP-TMR-SMX/CP-TMR(6)/BG-Alexa(488) and SiR-Halo (samples 5–7). For the quantification of TMR or SiR labeling, the ratio of Alexa (488)/SiR and TMR/SiR of the intracellular samples is calculated relative to the in vitro samples. The results of the labeling efficiency and the description of the samples run on the SDS-PAGE gel can be found in **Table b**. (**c**) Comparison of the endogenous SPR level of different cell lines by Western Blot. Western blot of SPR (28 kDa) and β-tubulin (50 kDa) as loading control with different cell lysates revealed by ECL. For each cell lysates, 20 μg total protein were loaded in each well. (**d**) Representation of the relative expression level of SPR in the different cell lines determined as integrated band intensity normalized to β-tubulin integrated intensity using the displayed blot. (**e**) The sensor dynamic range is maintained in lysate or in cells. The purified NADP-Snifit is added to a freshly prepared U2OS lysate (0.5 mg/mL protein) to a concentration of 50 nM. The measured TMR/SiR ratio of 1.6 corresponds to a NADPH/NADP$^+$ ratio of 11 in the whole-cell lysate (black line). The sensor was fully open by adding a saturating concentration of free ligand (2.5 mM sulfapyridine) and displays a TMR/SiR ratio of

4.5 (red line). To obtain the fully closed sensor in lysate, 10 mM NADP$^+$ was spiked to the lysate, resulting in a TMR/SiR ratio of 0.5 (blue line). A similar FRET ratio change can be observed for closed sensor in buffer. (f) Semi-stable U2OS cells expressing the nuclear localized NADP-Snifit were used to performed an intracellular sensor calibration. The cells plated on a 12-well plate poly-L-lysine coated coverslip were imaged in HBSS with a widefield microscope. After 2 min, 10 mM NADP$^+$ and 0.001% (w/w) digitonin prepared in HBSS was added to reach the sensor closed state. At 17 min, sulfapyridine was added to a saturating concentration (2 mM) to reach the sensor open state. The dynamic range measured with this widefield microscope was approximately of 8-fold similarly to lysate and buffer measurements.

DOI: https://doi.org/10.7554/eLife.32638.030

The following source data is available for figure :
**Appendix 1—figure 3—source data 1.**
DOI: https://doi.org/10.7554/eLife.32638.031
**Appendix 1—figure 3—source data 2.**
DOI: https://doi.org/10.7554/eLife.32638.032

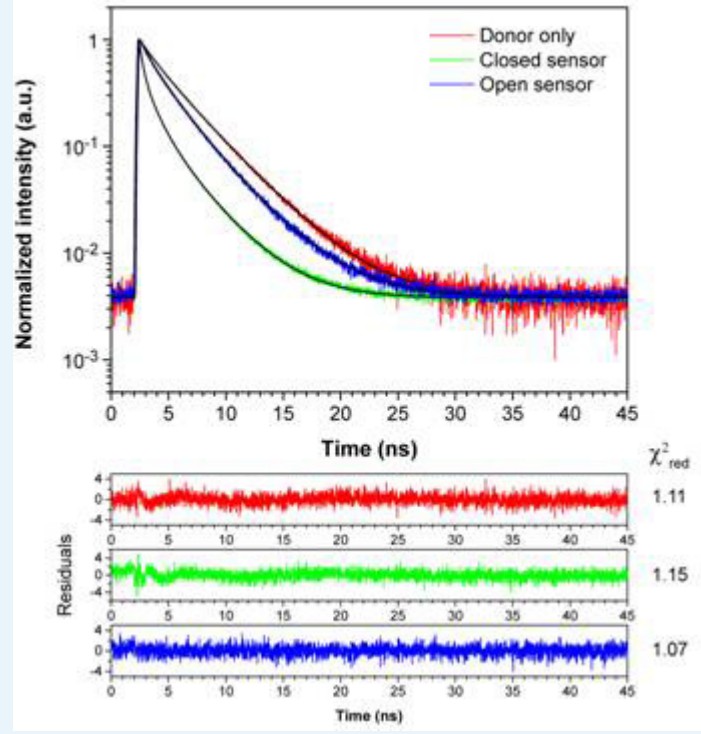

**Appendix 1—figure 4.** Fluorescence decays of the purified sensor measured by FLIM. Representative fluorescence decays of NADP-Snifit in buffer (50 mM HEPES, 150 mM NaCl, pH 7.5, 0.5 mg/mL BSA at 37 °C) measured by TCSPC-FLIM. The donor only sample corresponds to purified sensor labeled only with CP-TMR-SMX (red line). The fluorescence decay is fitted with a biexponential model ($\chi^2_{red}$ = 1.11), yielding an amplitude-weighted average lifetime $\langle\tau\rangle$ of 2.84 ns. The FRET samples are prepared with the addition of 1 mM NADP$^+$ (green line) or 2 mM sulfapyridine (blue line) in the aforementioned buffer to obtain, respectively the closed sensor with highest FRET efficiency ($E_{max}$) and the closed sensor with the lowest FRET efficiency ($E_{min}$). The fluorescence decays of FRET samples are best fitted with a 3rd-order exponential model. The closed and open sensor conformation yield $\langle\tau\rangle$ of 1.03 ($\chi^2_{red}$ = 1.15) and 2.4 ns ($\chi^2_{red}$ = 1.07), respectively. $E_{max}$ and $E_{min}$ correspond to 64% and 15% according to the following equation: $E(\%) = \left(1 - \frac{\tau_{DA}}{\tau_D}\right) \times 100$; where $\tau_{DA}$ is the lifetime of the FRET sample (donor + acceptor) and $\tau_D$ is the lifetime of the donor-only sample.

DOI: https://doi.org/10.7554/eLife.32638.033

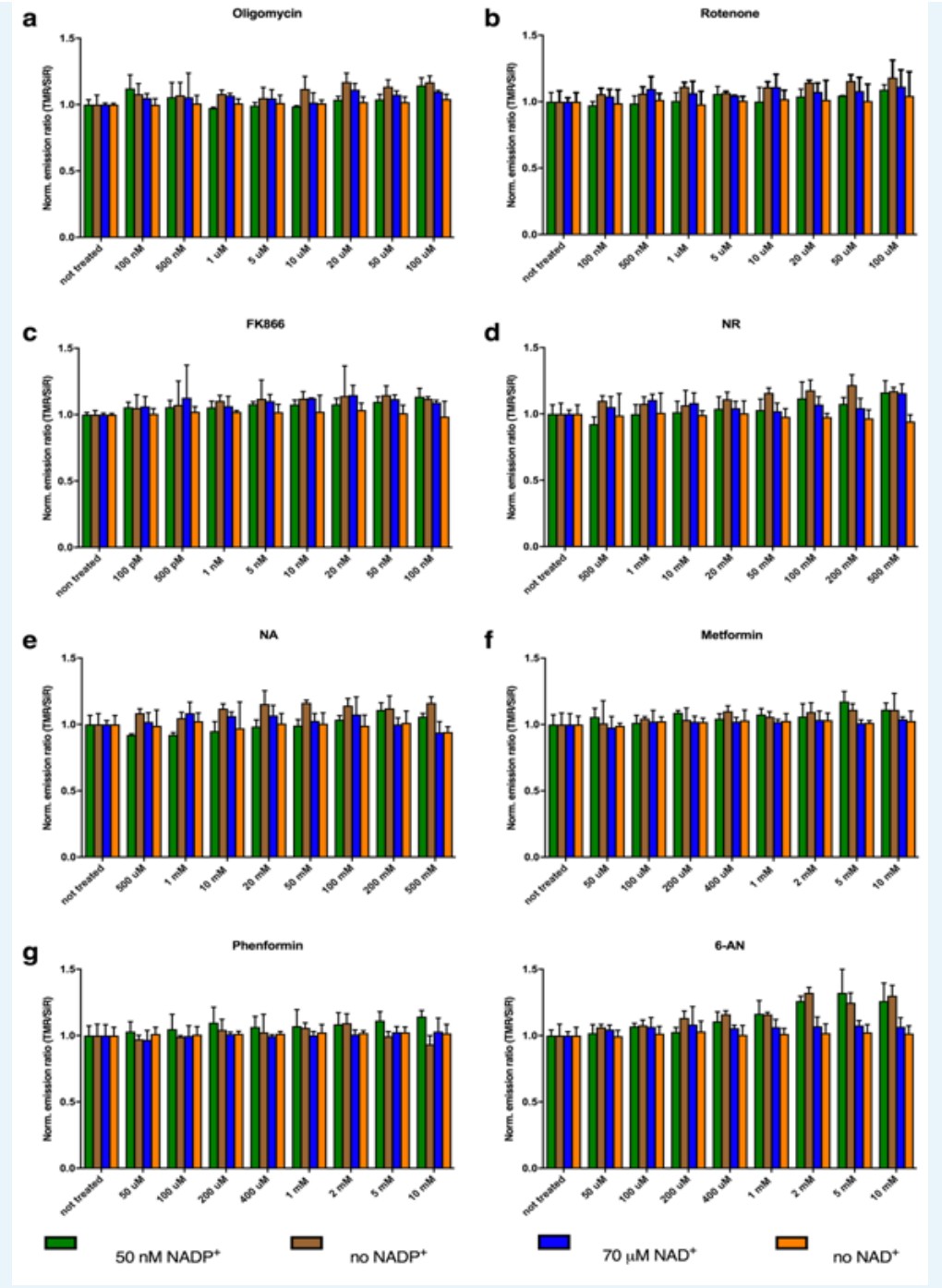

**Appendix 1—figure 5.** In vitro titration to assess potential interference of the investigated compounds on the sensor's performance. NAD(P)-Snifits were incubated with different concentrations of the compounds in the presence of the respective cofactor. Cofactor concentrations were set around the $c_{50}$ as perturbations on the sensor's performance should be most prominent in this range. The same experiments were performed in the absence of the cofactor to exclude that the sensors can be closed by the compounds. No significant perturbation of the sensor's performance was observed. (**a**) Oligomycin, (**b**) Rotenone, (**c**) FK866: (**E**)-N-[4-(1-benzoylpiperidin-4-yl)butyl]−3-(pyridine-3-yl)acrylamide, (**d**) NR: nicotinamide riboside, (**e**) NA: nicotinic acid, (**f**) Metformin, (**g**) Phenformin, (**h**) 6-AN. Three independent experiments were measured and the mean values including ± s.d. are shown.

DOI: https://doi.org/10.7554/eLife.32638.034

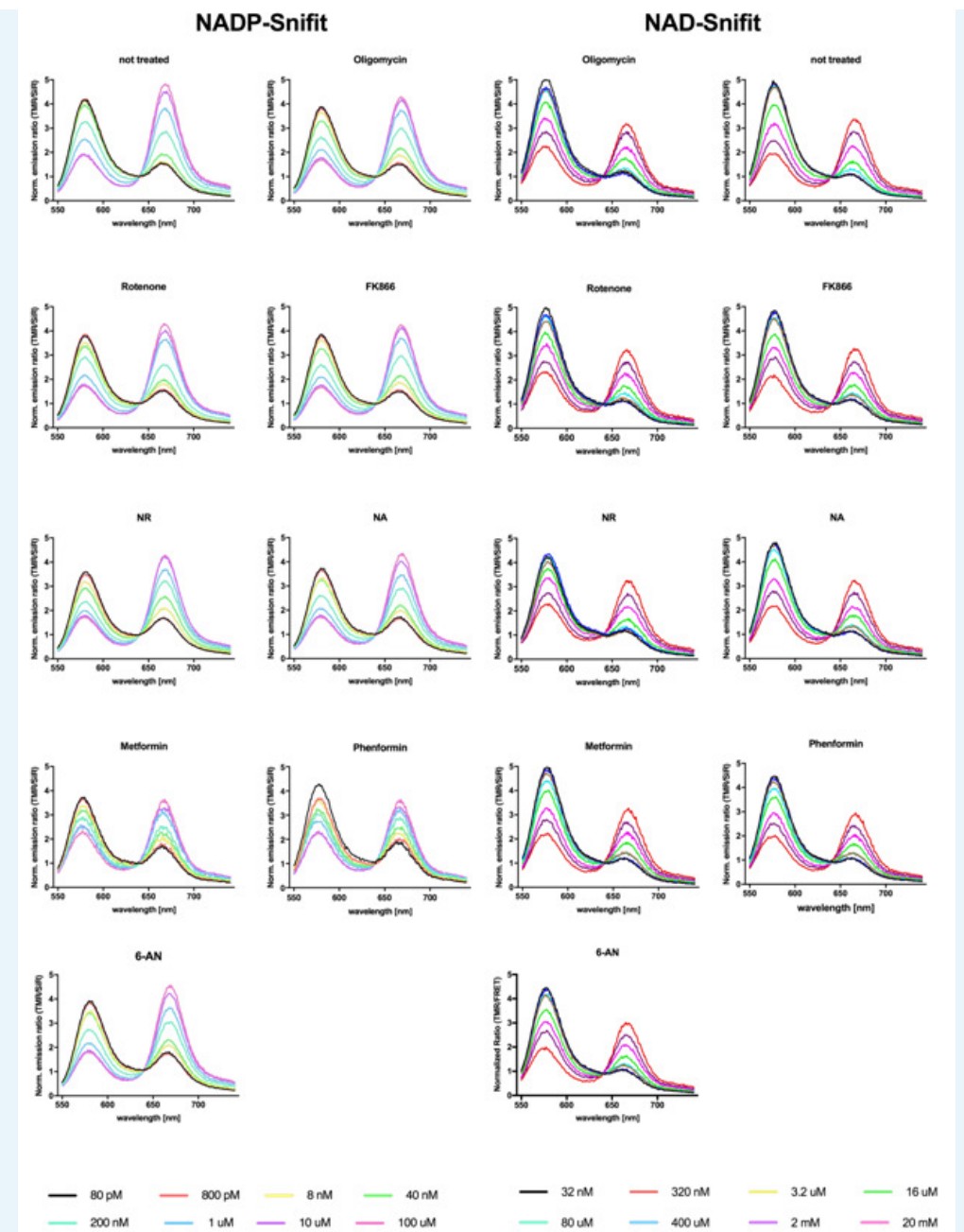

**Appendix 1—figure 6.** Emission spectra of the NAD(P)-Snifit titrates in the presence of investigated compounds. The emission spectra were recorded in the presence of the respective compound using the same conditions as for the FACS experiments. No significant alteration of the spectra was observed. Conditions: untreated control, 25 µM Oligomycin, 10 µM Rotenone, 100 nM FK866: (**E**)-N-[4-(1-benzoylpiperidin-4-yl)butyl]−3-(pyridine-3-yl)acrylamide, 10 mM NR: nicotinamide riboside, 1 mM NA: nicotinic acid, 1 mM Metformin, 1 mM Phenformin.

DOI: https://doi.org/10.7554/eLife.32638.035

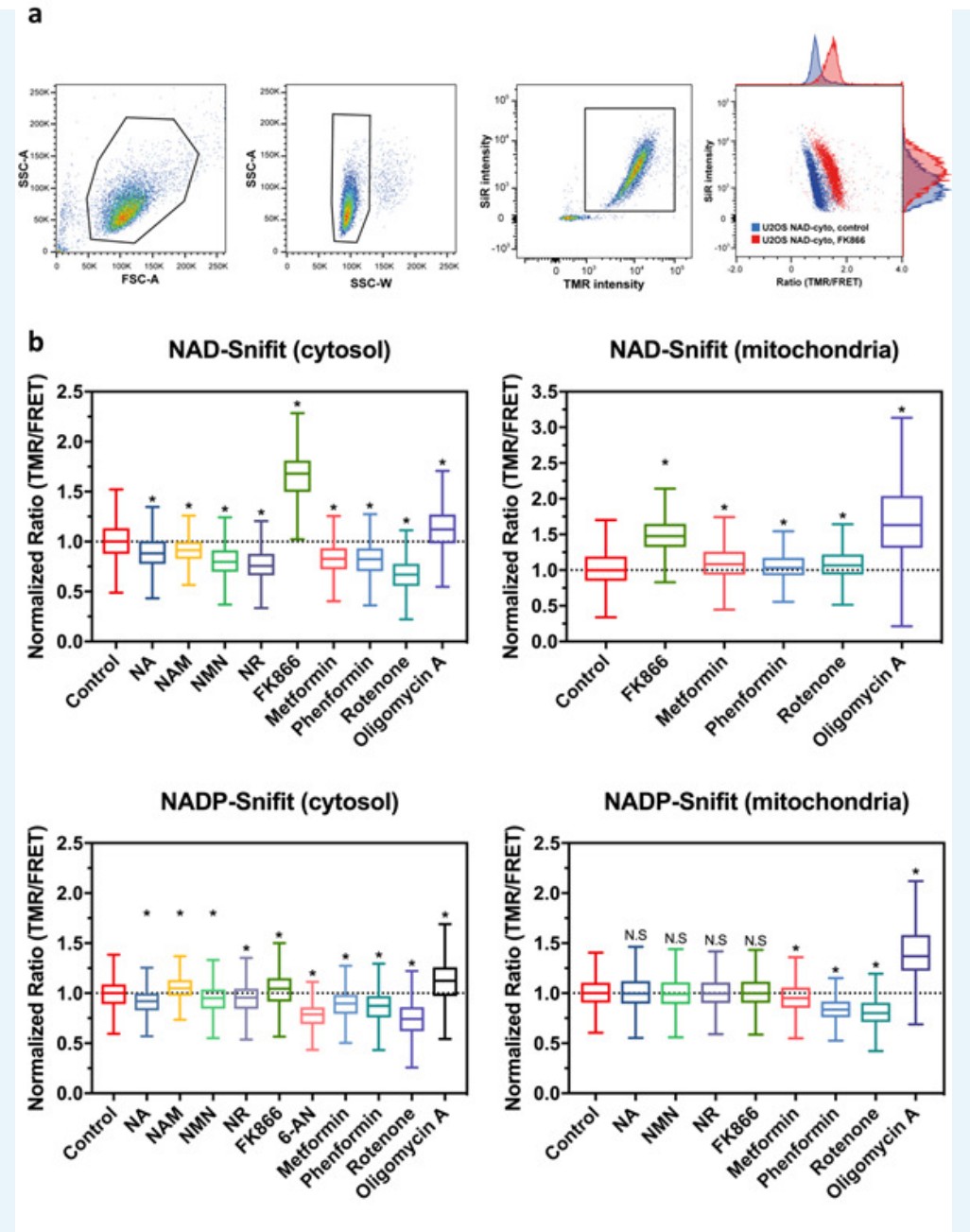

**Appendix 1—figure 7.** Monitoring NAD[+] and NADPH/NADP[+] by flow cytometry, related to *Figure 4* and *Table 2*. (**A**) Representative dot plots of the gating strategy. Live cells and singlets were gated by excluding dead cells and cellular debris (SSC-A vs FCS-A) and doublets or cell clumps (SSC-A vs SSC-W), respectively. Then, only the population of cells with sensor construct fully labeled with CP-TMR-SMX and SiR-Halo were considered for analysis. Example of FRET ratio (TMR/FRET) changes of U2OS cell populations treated for 24 hr with 100 nM of the non-competitive NAMPT inhibitor FK866 (red, significantly increasing the ratio TMR/FRET (i.e. decrease NAD[+] level)) compared to the untreated cells (control, blue) (n = 7000 gated cells per condition). (**B**) Flow cytometry data of cytosolic and mitochondrial NAD(P)-Snifits in U2OS cells after 24 hr incubations under the conditions specified. 1 mM NA, 10 mM NAM, 1 mM NR, 100 nM FK866, 1 mM 6-AN, 1 mM Metformin, 1 mM Phenformin, 10 μM Rotenone and 25 μM Oligomycin A. The results represent the sensor responses measured as FRET ratio (TMR/FRET) normalized to the untreated cell population. The Tukey-style box plots represent the 25th and 75th percentiles at the lower and upper box limits and the median as the middle

bar. The whiskers extend to ± 1.5 x IQR beyond the limits of the boxes, respectively. The position of the mean is indicated by a solid square. The data represent one data set for each condition (n = 2000–7000 events). * p<0.05 (Kruskal-Wallis with Dunn's post-hoc multiple comparison test with respect to control conditions), n.s. = not significant.

DOI: https://doi.org/10.7554/eLife.32638.036

The following source data is available for figure :
**Appendix 1—figure 7—source data 1.**
DOI: https://doi.org/10.7554/eLife.32638.037

**Appendix 1—table 1.** Properties of TMR and SiR substrates.

| Substrate name | Reaction rate constant ($M^{-1}$ $s^{-1}$) (± SD) | Excitation maximum (nm) | Emission maximum (nm) | Lifetime* (ns) |
|---|---|---|---|---|
| BG-TMR(6) | 114'706 (± 5082) | 555 | 577 | 2.2 |
| CP-TMR(6) | 109'278 (± 3338) | 555 | 577 | 2.2 |
| BG-TMR-SMX | 25'836 (± 2307) | 555 | 577 | 2.7 |
| CP-TMR-SMX | 38'847 (± 2307) | 555 | 577 | 2.7 |
| SiR-Halo | >250'000 [10] | 650 | 667 | 3.1 |

Values for the excitation/emission maxima and the lifetimes were measured with the protein-bound fluorophore.
*The lifetimes correspond to the amplitude-weighted average lifetime measured at 22 $\deg$C in HEPES buffer.

DOI: https://doi.org/10.7554/eLife.32638.038

**Appendix 1—table 2.** Intracellular NAD(P)-Snifit concentration.

| Localization | Concentration (μM) (± SD) | N cells |
|---|---|---|
| Cytosol | 1.6 (± 1.4) | 84 |
| Nucleus | 4.0 (± 3.3) | 51 |
| Mitochondria | 4.7 (± 1.3) | 49 |

Concentrations were determined in U2OS cells using a confocal fluorescence microscope by singly labeled the sensor construct with SiR-Halo and CP-TMR-SMX and comparing with the purified sensor calibration curves in buffer using identical microscope settings.

DOI: https://doi.org/10.7554/eLife.32638.039

**Appendix 1—table 3.** Quantification of the cytosolic free [NADPH]/[NADP$^+$] and [NAD$^+$] in different cell lines by TCSPC-FLIM.

| Cell lines | [NADPH]/[NADP$^+$] | [NAD$^+$] (μM) |
|---|---|---|
| U2OS | 55.8 ± 11.7 | 73.9 ± 7.1 |
| HEK293T | 21.6 ± 3.4 | 63.6 ± 4.5 |
| NIH/3T3[§] | 39.5 ± 12.4 | 44.6 ± 11.2 |
| HeLa | 75.0 ± 11.8 | 49.8 ± 2.4 |

DOI: https://doi.org/10.7554/eLife.32638.040

**Appendix 1—table 4.** Estimated free [NAD$^+$] and [NADPH]/[NADP$^+$] of pharmacologically treated U2OS cells measured by flow cytometry.

| | Free [NAD$^+$] (μM) | Free [NADPH]/[NADP$^+$] ratio |
|---|---|---|

*Appendix 1—table 4 continued*

| Compound | Free [NAD$^+$] (µM) | | Free [NADPH]/[NADP$^+$] ratio | |
|---|---|---|---|---|
| **Compound** | **Cytosol** | **Mitochondria** | **Cytosol** | **Mitochondria** |
| Control | 132 (± 29) | 96 (± 20) | 72 (± 8) | 120 (± 14) |
| 1 mM NA | 162 (± 33) | n.d. | 52 (± 6) | 119 (± 14)[†] |
| 10 mM NAM | 146 (± 30) | n.d. | 87 (± 10) | n.d. |
| 1 mM NMN | 198 (± 41) | n.d. | 59 (± 7) | 114 (± 14)[†] |
| 10 mM NR | 210 (± 43) | n.d. | 60 (± 7) | 120 (± 14)[†] |
| 100 nM FK866 | 42 (± 9) | 34 (± 7) | 88 (± 10) | 116 (± 14)[†] |
| 1 mM 6-AN | n.d. | n.d. | 35 (± 5) | n.d. |
| 1 mM Metformin | 168 (± 35) | 80 (± 17) | 49 (± 6) | 90 (± 10) |
| 1 mM Phenformin | 213 (± 45) | 72 (± 15) | 45 (± 6) | 54 (± 6) |
| 10 µM Rotenone | 300 (± 64) | 81 (± 17) | 29 (± 4) | 47 (± 6) |
| 25 µM Oligomycin A | 101 (± 21) | 24 (±5) | 121 (± 24) | open sensor* |

Values represent the mean estimated concentrations and ratios (± SD) of three independent measurements performed in triplicate. The TMR/FRET ratios were converted into concentration using **Equations 7 and 8**, where $R_{max}$ was determined in situ by incubating 10 min the cells with 2 mM sulfapyridine. $R_{min}$ was calculated from the in vitro maximum FRET ratio change $\Delta R_{max}$ ($R_{min} = R_{max}/\Delta R_{max}$). $c_{50}$ and $r_{50}$ were determined from in vitro titrations at 25 degC. Control: untreated cells (full growth medium with 25 mM glucose), NA: nicotinic acid, Nam: nicotinamide, NMN: nicotinamide mononucleotide, NR: nicotinamide riboside, FK866: (E)-N-[4-(1-benzoylpiperidin-4-yl)butyl]−3-(pyridin-3-yl)acrylamide, 6-AN: 6-aminonicotinamide.
*The sensor reached full opening with this treatment [NADPH]/[NADP$^+$] $\geq$ 300.
†The effect of the treatment is not statistically different compared to the control condition (p $\geq$ 0.05 using a two-tailed Student's $t$-test). n.d., not determined.

DOI: https://doi.org/10.7554/eLife.32638.041

**Appendix 1—table 5.** In vitro lifetime characterization of NADP-Snifit.

| Sample | <τ> (ns) ± SD | E (%) ± SD |
|---|---|---|
| Donor-only | 2.84 ± 0.01 | - |
| FRET (closed sensor) | 1.03 ± 0.01 | 63.9 ± 0.1 |
| FRET (open sensor) | 2.40 ± 0.01 | 15.3 ± 0.1 |

The 'donor only' sample represents the purified sensor singly labeled with CP-TMR-SMX. The FRET samples corresponding to the closed and open sensor state were prepared respectively with 1 mM NADP$^+$ and 2 mM sulfapyridine. The amplitude-weighted average lifetimes <τ> are represented as mean ± SD of triplicates. All samples were measured in buffer (50 mM HEPES, 150 mM NaCl, 0.5 mg/mL BSA, pH 7.5) at 37 degC. From the obtained lifetimes, the FRET efficiency (E) of the closed and open sensor was calculated.

DOI: https://doi.org/10.7554/eLife.32638.042

**Appendix 1—table 6.** Determination of FRET efficiency in U2OS cells.

| | | <τ> (ns) ± SD | | | E (%) ± SD (Basal) | E$_{min}$ (%) ± SD (2 mM SPY) |
|---|---|---|---|---|---|---|
| **Sensors** | **Localization** | **Donor only** | **FRET, basal** | **FRET, 2 mM SPY** | | |

*Appendix 1—table 6 continued on next page*

*Appendix 1—table 6 continued*

| Sensors | Localization | <τ> (ns) ± SD | | | E (%) ± SD (Basal) | E$_{min}$ (%) ± SD (2 mM SPY) |
|---|---|---|---|---|---|---|
| | | Donor only | FRET, basal | FRET, 2 mM SPY | | |
| NADP-Sni-fit | Cytosol | 2.80 ± 0.02 | 2.12 ± 0.06 | 2.35 ± 0.04 | 24.2 ± 0.7 | 16.0 ± 0.3 |
| | Nucleus | 2.69 ± 0.03 | 1.98 ± 0.04 | 2.28 ± 0.04 | 26.3 ± 0.6 | 15.5 ± 0.4 |
| | Mitochondria | 2.55 ± 0.05 | 2.08 ± 0.03 | 2.16 ± 0.02 | 18.2 ± 0.4 | 15.2 ± 0.3 |
| NAD-Snifit | Cytosol | 2.89 ± 0.04 | 2.19 ± 0.02 | 2.42 ± 0.02 | 24.3 ± 0.4 | 16.3 ± 0.3 |
| | Nucleus | 2.66 ± 0.03 | 2.11 ± 0.04 | 2.49 ± 0.06 | 20.6 ± 0.4 | 6.5 ± 0.2 |
| | Mitochondria | 2.63 ± 0.03 | 2.02 ± 0.02 | 2.30 ± 0.06 | 23.0 ± 0.3 | 12.2 ± 0.3 |

The data represent the amplitude-weighted average lifetime <τ> as mean ± SD (N = 10) measured in living U2OS cells in full growth medium (DMEM +10% FBS) at 37 degC. The 'donor-only' sample was obtained by single labeling of the sensor constructs with CP-TMR-SMX. The FRET samples are labeled with both CP-TMR-SMX and SiR-Halo. The cells labeled with both fluorophores were first measured without treatment to obtain their basal fluorescence lifetime, then the same cells were measured again after the treatment with 2 mM sulfapyridine (SPY) to obtain the fully sensor open state. The correlated FRET efficiencies (E) were calculated for each conditions.

DOI: https://doi.org/10.7554/eLife.32638.043

