## [Decision Letter]

Thank you for submitting your article "Semisynthetic biosensors for mapping cellular concentrations of nicotinamide adenine dinucleotides" for consideration by *eLife*. Your article has been reviewed by three peer reviewers, one of whom, Yamuna Krishnan is a member of our Board of Reviewing Editors and the evaluation has been overseen by Michael Marletta as the Senior Editor. The following individuals involved in review of your submission have agreed to reveal their identity: Ralph Mazitschek (Reviewer #2); David Sinclair (Reviewer #3).

The reviewers have discussed the reviews with one another and the Reviewing Editor has drafted this decision to help you prepare a revised submission.

Summary:

The authors developed a 'semisynthetic' fluorescent biosensors with large dynamic ranges to measure both free NAD^+^ and the ratio of NADPH/NADP^+^ in live cells. They also report the use of these sensors to map NADPH/NADP^+^ ratios in subcellular compartments and in assessing the effects of caloric restriction on redox state.

Essential revisions:

The reviewers were enthusiastic about the performance of the sensors and their potential to provide useful biological insights. They suggested that the revised manuscript pay special attention to the following major issues.

1) Provide a side-by-side comparison of their Snifit sensor vs SoNar and iNap sensors quantified by flow cytometry to support the assertion that their system is more sensitive and amenable for HTS.

2) Measure FRET efficiencies in the presence of various concentrations of resveratrol or oligomycin in vitro, to determine if polyphenolic compounds do change the sensitivity of the assay, especially at high concentrations. For HTS to be feasible, other compounds should not interfere with the sensor's performance.

3.) Show measurements of NAD-Snifit output in cells using various concentrations of resveratrol as 100 μm is likely to be toxic. Typically, experiments have been carried out at 25 μm and in some circumstances cells begin to die even at 12.5 μm (see Gomes et al., 2013).

4) Clarify the role of NAD^+^ measurements. The reviewers' enthusiasm was focused on the NADP^+^/NADPH ratio measurements, and it's not clear that this requires NAD^+^ measurement. Would the manuscript be clearer if NAD^+^ were omitted?

[Editors’ note: this article was subsequently rejected after discussions between the reviewers, but the authors were invited to resubmit after an appeal against the decision.]

Thank you for resubmitting your work entitled "Semisynthetic biosensors for mapping cellular concentrations of nicotinamide adenine dinucleotides" for consideration by *eLife*. Your revised article has been reviewed by one external peer reviewer, and the evaluation has been overseen by a Reviewing Editor and a Senior Editor. The reviewers have opted to remain anonymous.

Our decision has been reached after consultation between the reviewers. Based on these discussions and the individual review below, we regret to inform you that your work will not be considered further for publication in *eLife*.

We appreciate the authors' efforts in responding to the reviews. However, the response to the most important request – performance characteristics that would establish the importance of the Snifit sensor's capabilities in relation to those of the existing SoNar and iNap sensors – fell short. The response that the Snifit sensor was 'complementary' to the other sensors really didn't provide a compelling description of the complementarity.

Reviewer #1:

Unfortunately, the revised manuscript does little to address the original concerns of the reviewers.

The authors state that the rationale provided for not providing a side-by-side comparison with SoNar or iNAP is that they are complementary and not competing as authors stated earlier. However, the authors have not vetted the model of their present sensor rigorously by showing that the amount of high FRET, non-NADPH bound and non-NADP^+^ bound sensor is negligible under physiological conditions for a complete range of NADPH/NADP^+^ ratios. No reasons have been provided for why a different mode of analysis was not possible, even if not done by the authors.

The toxicity of resveratrol and the removal of the section on calorific restriction basically neuters the paper in terms of significance content by removing the application of the NAD^+^ sensor. This leaves the reader even more confused on the significance of this particular NAD^+^ sensor – there are already good sensors for NAD^+^ (Zhao 2015, Cambronne 2016), and detracts from the novelty of the NADPH/NADP sensor.

I see that this is still being considered as a research article, though the recommendation last time was a Tools/resources article – provided the authors address all the concerns. As it stands, the manuscript does not reach the significance requirements for a research article in *eLife*, neither have the authors fully addressed some major concerns raised by multiple reviewers needed for a Tool or resource.

---

## [Author Response]

Essential revisions:The reviewers were enthusiastic about the performance of the sensors and their potential to provide useful biological insights. They suggested that the revised manuscript pay special attention to the following major issues.1) Provide a side-by-side comparison of their Snifit sensor vs SoNar and iNap sensors quantified by flow cytometry to support the assertion that their system is more sensitive and amenable for HTS.

The sensors SoNar and iNAP measure NADH/NAD^+^ ratios and free NADPH concentrations, respectively. Both sensors have been successfully used in flow cytometry applications. NAD-Snifit and NADP-Snifit measure free NAD^+^ and NADPH/NADP^+^, respectively. A side-by-side comparison of these sensors would be interesting as it would reveal how changes in absolute free NAD(P) concentrations are related to changes in redox ratios. However, as the different sensors measure different analytes, such measurements could not be used to compare their performance. Furthermore, we did not want to imply that our sensors are more sensitive and amenable to HTS than SoNar and iNAP but rather are of the opinion that they are complementary, both with respect to analytes as well as colors. We have modified the manuscript accordingly to stress that point (subsection “Pharmacological alteration of cellular metabolism”).

For the reasons listed above we would like to ask the reviewers to accept our decision not to perform these experiments.

2) Measure FRET efficiencies in the presence of various concentrations of resveratrol or oligomycin in vitro, to determine if polyphenolic compounds do change the sensitivity of the assay, especially at high concentrations. For HTS to be feasible, other compounds should not interfere with the sensor's performance.

We have characterized the effect of all molecules used in this study on the in vitro FRET efficiencies. These data are summarized in the new Supplementary Figures 5 and 6. At physiologically relevant concentrations, none of the compounds used in the flow cytometry assays effects sensor performance.

3.) Show measurements of NAD-Snifit output in cells using various concentrations of resveratrol as 100 μm is likely to be toxic. Typically, experiments have been carried out at 25 μm and in some circumstances cells begin to die even at 12.5 μm (see Gomes et al., 2013).

We are very grateful to the reviewers for this suggestion. We have now measured the effect of various resveratrol concentrations on U2OS cell viability and indeed have observed that already at 10 µM of resveratrol an effect on cell viability could be observed. The data of the toxicity measurements are summarized at the end of this letter. This observed toxicity complicates an interpretation of the effect of resveratrol on free NAD^+^ and NADPH/NADP^+^. We have therefore decided to remove the resveratrol measurements and the discussion on caloric restriction from the manuscript. We believe that focusing the manuscript on (i) the technical aspects of this work and (ii) the NADPH/NADP^+^ measurements should also address comment No 4 of the reviewers (see below).

4) Clarify the role of NAD^+^ measurements. The reviewers' enthusiasm was focused on the NADP^+^/NADPH ratio measurements, and it's not clear that this requires NAD^+^ measurement. Would the manuscript be clearer if NAD^+^ were omitted?

We have tried to address this comment by removing the part of the manuscript describing the experiments with resveratrol and caloric restriction, as these mainly deal with increases in free cytosolic NAD^+^. However, we are of the opinion that a description of the NAD^+^ sensor should remain a part of this manuscript as it (i) highlights the modular nature of the sensor design and (ii) introduces a valuable resource to the scientific community.

[Editors’ note: the author responses to the re-review follow.]

We appreciate the authors' efforts in responding to the reviews. However, the response to the most important request – performance characteristics that would establish the importance of the Snifit sensor's capabilities in relation to those of the existing SoNar and iNap sensors – fell short. The response that the Snifit sensor was 'complementary' to the other sensors really didn't provide a compelling description of the complementarity.Reviewer #1:Unfortunately, the revised manuscript does little to address the original concerns of the reviewers.The authors state that the rationale provided for not providing a side-by-side comparison with SoNar or iNAP is that they are complementary and not competing as authors stated earlier. However, the authors have not vetted the model of their present sensor rigorously by showing that the amount of high FRET, non-NADPH bound and non-NADP^+^ bound sensor is negligible under physiological conditions for a complete range of NADPH/NADP^+^ ratios. No reasons have been provided for why a different mode of analysis was not possible, even if not done by the authors.The toxicity of resveratrol and the removal of the section on calorific restriction basically neuters the paper in terms of significance content by removing the application of the NAD^+^ sensor. This leaves the reader even more confused on the significance of this particular NAD^+^ sensor – there are already good sensors for NAD^+^ (Zhao 2015, Cambronne 2016), and detracts from the novelty of the NADPH/NADP sensor.I see that this is still being considered as a research article, though the recommendation last time was a Tools/resources article – provided the authors address all the concerns. As it stands, the manuscript does not reach the significance requirements for a research article in eLife, neither have the authors fully addressed some major concerns raised by multiple reviewers needed for a Tool or resource.

Thank you for overviewing the review process of our manuscript. I am also grateful to the reviewers for taking the time to look at our revised manuscript. Naturally, I am very disappointed by the outcome. I understand that most authors of rejected manuscripts disagree with editorial decisions. Nevertheless, I am convinced that in this case the reasons for rejecting our work are incorrect for the following reasons and I would like to ask you to reconsider your decision:

1) As you will recall, the initial reviews were quite supportive. The main reason for the rejection of the revised manuscript was that we decided not to do the side-by-side comparison of our two sensors with two previously published sensors (SoNAr and iNap) that measure NADH/NAD^+^ (in the case of SoNAr) and NADPH (in the case of iNap). Our sensors measure NADPH/NADP^+^ and NAD^+^. We wrote in our point-by-point response that we decided not to do the requested comparisons as one must not compare a sensor that measures a ratio of two analytes with a sensor that measures an absolute concentration of one of these two analytes. I still believe that fulfilling this request would not provide useful information on our sensors; one should not compare apples with pears. In the transmitted reviewer comment it is stated that “No reasons have been provided for why a different mode of analysis was not possible, even if not done by the authors”. As stated above and in our previously submitted point-by-point response, the reason is that a direct comparison of sensors that measure different analytes is not meaningful. This is also why we consider the different sensors as “complementary” as they provide information on different NAD(P) concentrations and redox ratios in cells.

2) In the transmitted comments from the reviewers it is also stated that: “….However, the authors have not vetted the model of their present sensor rigorously by showing that the amount of high FRET, non-NADPH bound and non-NADP^+^ bound sensor is negligible under physiological conditions for a complete range of NADPH/NADP^+^ ratios.”

This criticism was not raised in the original response we received. However, we have stated already in the first version of our manuscript that for both sensors the observed FRET ratios are independent of expression levels of the sensor (as measured by fluorescence intensities). Please see the Results section of the revised manuscript. I agree with the reviewers that this aspect of our work could have been explained better. We would be happy to do this in a revised version of our manuscript by including additional data that further demonstrate that the measured FRET ratios are independent of sensor concentrations. I should also mention that in contrary to what is stated by the reviewers, our sensors show very little FRET when not bound to cofactor.

3) Furthermore, in the reviewer comments it is stated that our manuscript " …is still being considered as a research article, though the recommendation last time was a Tools/resources article ". This recommendation was not communicated to us. However, I believe that this is a very good suggestion of the reviewers as the revised version of the manuscript focuses much more on the design and characterization of the sensors. We certainly would be happy to submit a manuscript on our sensors to the “Tools” section of *eLife*.

4) Finally, the reviewers state that “there are already good sensors for NAD^+^ (Zhao 2015, Cambronne 2016), and this detracts from the novelty of the NADPH/NADP sensor." The work of Zhao introduces SoNar, a sensor for measuring NADH/NAD^+^ ratios but not absolute concentrations of NAD^+^. The work of Cambronne indeed introduces a sensor for NAD^+^, but as we state in our paper this sensor has a number of shortcomings (please see the Introduction): „While being the first sensor able to measure free, compartmentalized NAD^+^, it only shows a modest two-fold dynamic range and requires excitation at 405 and 488 nm. Furthermore, the pH sensitivity of the fluorescence signal of the sensor between pH 7.4 to 8 is comparable to its dynamic range.” There are thus good reasons to introduce our NAD^+^ sensor to the scientific community and we actually already have shared both plasmids and reagents for this sensor with a number of other labs who are interested in using it.